# The fungal collaboration gradient drives root trait distribution and ecosystem processes in tropical montane forests

Mateus Dantas de Paula[1], Tatiana Reichert[2], Laynara F.Lugli[2], Erica McGale[3], Kerstin Pierick[7], João P. Darela-Filho[2], Liam Langan[1], Jürgen Homeier[4,5], Anja Rammig[2], Thomas Hickler[1,6]

[1]Senckenberg Biodiversity and Climate Research Centre (SBiK-F), Georg-Voigt-Straße 14-16, 60325, Frankfurt am Main, Germany
[2]Professorship for Land Surface-Atmosphere Interactions, Technical University of Munich, Hans-Carl-v.-Carlowitz-Platz, 2, Freising, 85354, Bavaria, Germany
[3]Department of Ecology and Evolution (DEE), University of Lausanne, 1015 Lausanne, Switzerland
[4]HAWK University of Applied Sciences and Arts, Faculty of Resource Management, Daimlerstraße 2, 37075 Goettingen, Germany
[5]University of Goettingen, Plant Ecology and Ecosystems Research, Untere Karspüle 2, 37073 Goettingen, Germany
[6]Department of Physical Geography, Geosciences, Johann Wolfgang Goethe University of Frankfurt, Frankfurt, Germany
[7]University of Goettingen, Spatial Structures and Digitization of Forests/Silviculture and Forest Ecology of the Temperate Zones, Büsgenweg 1, 37077 Göttingen, Germany

*Correspondence to*: Mateus Dantas de Paula (mateus.dantas@senckenberg.de)

1. **Abstract.** Plant roots have a large diversity of form and function, which is also related to their degree of mycorrhizal association. This is known as the fungal collaboration gradient, where thinner roots acquire resources by themselves and thicker roots depend on mycorrhizas. In this study, we, for the first time, implement the fungal collaboration gradient in a trait-based Dynamic Vegetation Model (DVM, LPJ-GUESS-NTD). We test if the DVM can predict fine-root trait distributions, and estimate the effects of arbuscular mycorrhiza fungi (AMF)-mediated nutrient uptake on ecosystem processes along an elevation gradient in a tropical montane forest in southern Ecuador. The model reproduces the observed fine-root traits specific root length (SRL) and AMF colonization along the elevation gradient, which ranges from low AMF colonization at 1,000 m (25%) to high AMF colonization at 3,000 m (61%). When AMF-mediated nutrient uptake is deactivated site average biomass values are reduced by up to 80%. Accounting for AMF-related belowground traits also affects simulated community leaf traits, suggesting linkages between below- and aboveground traits by AMF promotion of more acquisitive leaf traits. In addition, deactivation of AMF uptake reduced simulated soil C stocks by up to 68%. The model suggests that the collaboration gradient has a substantial influence on vegetation diversity and functioning as well as soil carbon in the study system. We thus advocate more explicit treatment of fine-root traits and mycorrhizae in DVMs. The model scheme here is based on general trade-offs and could be implemented in other DVMs and be tested for other study regions.

## 2. Introduction

Belowground processes are becoming an increasingly important research topic in ecology, due in part to their overwhelming role in regulating biogeochemical cycles (Beillouin *et al.*, 2023). Soils at depths of up to 200 cm store an estimated 2400 Pg C globally, highlighting their importance in the carbon cycle (Batjes, 1996). This is nearly nine times the amount stored in global forests (Santoro *et al.*, 2021). Fine roots, the plant's interface with soil (as opposed to coarse roots, which have a more structural role), play an important role in driving ecosystem processes (Bardgett *et al.*, 2014; Weigelt *et al.*, 2021). In the

tropics, fine roots store up to 50 Mg C ha$^{-1}$ (Jackson *et al.*, 1997) with a productivity of around 6 Mg C ha$^{-1}$ y$^{-1}$ (Finér *et al.*, 2011), and are a major input to soil C stocks (Rasse *et al.*, 2005).

Fine-root morphological traits play a critical role in nutrient and water uptake. These traits may also shape species coexistence, and thus community composition in specific environments (Nie *et al.*, 2013). Biotic interactions of fine roots and other organisms are widespread, and conservative estimates suggest that ca. 20,000 plant species could rely on soil biota

to persist in natural, and especially nutrient-poor, environments (Van Der Heijden *et al.*, 2008). For instance, mycorrhizae support nutrient acquisition in exchange for C (Bardgett *et al.*, 2014; Bennett & Groten, 2022), with many plant species exhibiting fine-root traits that maximize this interaction, such as increased diameter and cortex area (Gu *et al.*, 2014; Kong *et al.*, 2014; Valverde-Barrantes *et al.*, 2021; Shi *et al.*, 2023). Mycorrhizal fungi are a major player in the global C cycle, drawing an average of 3-13%, but up to 50% of the plant partner's net primary production (NPP) (Hawkins *et al.*, 2023).

This means that fine-root traits may influence larger-scale ecosystem processes.

By analyzing the co-occurrence of plant traits, researchers have identified relationships between them and developed the concept of the global spectrum of plant form and function (Díaz *et al.*, 2015; Guerrero-Ramírez *et al.*, 2020; Kattge *et al.*, 2020)This includes the conservation trade-off axis, where morphological and stoichiometric traits (e.g. leaf C:N, tissue densities) are related to high productivity or high longevity strategies (Chave *et al.*, 2009; Wright *et al.*, 2013; Díaz *et al.*,

2015). This concept highlights how trade-offs in physiological and morphological traits influence species coexistence (Shipley *et al.*, 2006).. Advances in belowground trait measurements have enabled researchers to synthesize fine-root traits within the global spectrum of plant form and function (Weemstra *et al.*, 2016, 2022; Weigelt *et al.*, 2021). Fine-root stoichiometry traits and root tissue density were also observed to produce such a conservation gradient, however other fine-root morphological traits such as diameter and specific root length did not seem to align with the existing conservation axis

(Carmona *et al.*, 2021).

A promising plant form and function trait spectrum framework for understanding fine-root morphological trait variation is the "fungal collaboration gradient", one of the main components of the Root Economics Space framework proposed by Bergmann *et al.*, (2020). In addition to nutrient concentrations in their tissues produced from their own metabolism, plant fine roots and their morphological traits significantly affect their capacity to forage additional nutrients and water. For

instance, thinner and longer fine roots have a higher total soil absorptive area per mass when compared to thicker and shorter

fine roots (McCormack *et al.*, 2015). The conserved presence of thicker fine roots in several plant species, however, shows that individuals with this trait may also have benefits. Thicker, large-diameter fine roots that are usually associated with greater cortex:stele ratios have a higher anatomical capacity to be colonized by mycorrhizas, which can mediate the transfer of nutrients and water (Gu *et al.*, 2014; Kong *et al.*, 2014; Valverde-Barrantes *et al.*, 2021; Shi *et al.*, 2023). Although plants must transfer carbon to fungi as part of their partnership, the fungi's extensive hyphae networks can significantly boost nutrient and water absorption. This collaboration offers thicker rooted plants an alternative strategy to relying solely on roots for uptake of water and nutrients (Kakouridis *et al.*, 2022). A trade-off emerges between a cheaper "do-it-yourself" strategy, with fine roots of high specific root length (SRL) that provide sufficient absorptive area without the need for fungal collaboration; or "outsourcing" to the mycorrhizae partner, whereby a plant with fine roots of relatively thick dimeter and low surface area "pays" (i.e. transfers) carbohydrates to the fungus to benefit from the high absorptive capacities of hyphae (Bergmann *et al.*, 2020). A spectrum of strategies thus emerges: at one end, plants invest heavily in fine-root traits that enhance independent nutrient uptake, while at the other end they rely more on fungal partners to exchange nutrients for carbon.

Empirically-based concepts connecting patterns to processes such as the spectrum of form and function (Díaz *et al.*, 2015) are very useful for dynamic vegetation models (DVM), which depend on generalized representations of ecology. DVMs are software simulators of plant community, physiology and edaphic processes, used to conduct experiments which are difficult or inviable to produce in field settings, such as characterizations over large temporal and spatial scales from complex treatments (Prentice *et al.*, 2004; Sitch *et al.*, 2008; Quillet *et al.*, 2009). Aboveground plant trait variation has been implemented in DVMs using the 'leaf economics spectrum,' a framework that classifies leaves based on a trade-off between rapid growth and resource conservation (Wright *et al.*, 2004). Including such frameworks in DVMs has provided insights into nutrient dynamics and community resilience (i.e., Sakschewski *et al.*, 2015, 2016; Dantas de Paula *et al.*, 2021). By contrast, belowground processes, despite their critical ecological roles, remain underrepresented in these models (Langan *et al.*, 2017; Sakschewski *et al.*, 2021). Particularly, despite mycorrhizas being fundamental to plant growth, not many attempts have been made to explicitly include them in DVMs (Dantas de Paula *et al.*, 2019, 2021; He *et al.*, 2021; Kou-Giesbrecht *et al.*, 2021; Thurner *et al.*, 2023). None of these approaches was using trait-based models, meaning that the coexistence of species with lower and higher mycorrhiza colonization rates was not present, and results were highly parameter-dependent. Since data on mycorrhiza physiology and structure are scarce and complex to measure (Godbold *et al.*, 2006; Van Der Heijden *et al.*, 2008), these models may be producing results that are unrealistic. Nevertheless, including  variation in fine-root traits and mycorrhizal colonization in DVMs holds the potential to improve  predictions of vegetation response to climate change, as was the case with leaf and wood traits (Sakschewski *et al.*, 2016).Due to their increased absorption capabilities, mycorrhiza can effectively support plant survival in harsher environments, increasing resilience to disturbances such as droughts (Das & Sarkar, 2024).

Here, we derived parameters from the literature and field measurements to implement fine-root trait variations and the fungal collaboration gradient (FCG) in a trait-based DVM (LPJ-GUESS-NTD). This DVM includes N and P cycles, the latter of which is particularly important for tropical forests (Du *et al.*, 2020). Previously, an LPJ-GUESS-NTD version that included aboveground trait diversity, N and P cycling, and a simple mycorrhizal representation was applied in a well-researched elevation gradient in tropical mountane forest (TMF) biodiversity hotspot sites in the southern Ecuadorian Andes. The model reproduced an observed gradient of aboveground vegetation traits and C processes along the elevation gradient, showing that nutrient dynamics might play a very important role in vegetation changes and C cycling along this gradient (Dantas de Paula *et al.*, 2021). However, variation in fine-root traits was not accounted for, and mycorrhizal symbioses were highly simplified and prescribed. Such an approach can only yield limited site-specific insights about belowground traits and mycorrhiza. To address these gaps, this study develops a dynamic approach that integrates detailed fine-root trait and fungal interaction data into DVMs for broader ecological applications. It builds upon more detailed ecophysiological processes whose understanding is, as of yet, generalizable for DVM applications but which could be essential for linking above- and belowground plant traits. We applied the new model version to the Andes TMF and tested it against field observations of fine root and Arbuscular mycorrhizal fungi (AMF) measurements along the altitudinal gradient (Homeier & Leuschner, 2021; Pierick *et al.*, 2021). This local setting was chosen not only to build on previous results but also because we could apply the FCG using only one mycorrhiza type (AMF). We aim here to test with our model implementation the general hypothesis that the FCG is an important factor behind the observed fine-root trait distribution, forest biomass and productivity. , More specifically, we hypothesize that (1) in line with the mycorrhizal colonization gradient and field measurements of fine-root traits, as available nutrients decrease with elevation, simulated community average values of SRL decrease, diameters increase and colonization rates by AMF increase when the FCG is active. Next, we removed the mycorrhizal fungi in a simulated exclusion experiment, and (2)expect that in the absence of AMF, plant biomass and productivity would differ from observations, and SRL between the different sites of the elevation gradient would be similar. In other words, this latter result would imply that AMF drives morphological fine-root diversity.

## 3. Materials and Methods

### 3.1. Site description

To test our hypotheses and to drive and evaluate our simulations, we used site-level data from three different elevations (1,000, 2,000 and 3,000 m a.s.l.) in the Cordillera Real on the eastern range of the south Ecuadorian Andes. These three sites harbour old-growth forests and are located within or close to the Podocarpus National Park, where a wealth of biotic and abiotic measurements have been carried out since the early 2000's (Beck *et al.*, 2008; Bendix *et al.*, 2013, 2021). The terrain is predominantly steep and rugged, characterized by ridges and valleys. The primary parent materials for soil formation include Paleozoic metamorphosed schists and sandstones, interspersed with quartz veins. Cambisols are the most prevalent soil type at lower elevations, while Planosols and Histosols dominate at higher elevations around 2,450 meters (Wilcke *et al.*,

2008). The main recurring disturbance in the area are landslides (Wilcke *et al.*, 2003). The annual mean temperature among elevations decreases from around 20° C (1000 m) to ~10° C at (3000 m), and annual precipitation increases from 2230 mm (1000 m), and 1950 mm (2000 m), to 4500 mm (3000 m) (Moser *et al.*, 2007; Dietrich *et al.*, 2016; Bendix *et al.*, 2021). Due to several abiotic and biotic factors, of which temperature is the most relevant, decomposition and mineralization of organic matter declines with elevation along this range, resulting in an increase of the thickness of the organic layer from around 5 cm in the 1,000 m site to more than 50 cm in the 3,000 m site. Conversely, inorganic nutrient stocks, studied as plants' main nutrition source, decrease with elevation. In particular, nitrogen (both inorganic forms N; $Ni = NH_4\text{-}N + NO_3\text{-}N$) stocks vary from 5.14 kg Ni ha$^{-1}$ at 1,000 m, 13.02 kg Ni ha$^{-1}$ at 2,000 m and 0.64 kg Ni ha$^{-1}$ at 3,000 m on average (Velescu & Wilcke, 2020; Dantas de Paula *et al.*, 2021). Bray extractable inorganic phosphorus (Pi) measurements show no significant differences between elevation sites (Dietrich *et al.*, 2016), suggesting that P limitation may play a minor or co-limiting role to N in this gradient (Homeier *et al.*, 2012). Nutrient availability and the resulting limitation to plant growth along the elevation gradient are considered an important driver of the diversity of ecosystem processes and species functional traits in this region: for example, with increasing elevation, forest productivity and biomass stocks decrease (Homeier & Leuschner, 2021), and the community trait distribution becomes more conservative (Homeier *et al.*, 2021; Pierick *et al.*, 2023, 2024) (e.g. lower specific leaf area, lower fine-root nutrients).

Extensive floristic inventories in the area allowed the distinction of three main forest types: The studied forest types are classified as evergreen premontane forest (1000 m), evergreen lower montane forest (2000 m), and evergreen upper montane forest (3000 m); floristic composition changes rapidly with elevation, as most tree species in the study area are only found in narrow elevational belts (Homeier *et al.*, 2008). The evergreen premontane rain forest at the lowermost study site reaches 40 m in height with common tree families being Fabaceae, Moraceae, Myristicaceae, Rubiaceae and Sapotaceae. It is replaced at 1300–2100 m by smaller-statured lower montane rain forest with Euphorbiaceae, Lauraceae, Melastomataceae and Rubiaceae as characteristic tree families, and above 2100 m by upper montane rain forest with a canopy height rarely exceeding 8–10 m. Common tree families of the latter forest type are Aquifoliaceae, Clusiaceae, Cunoniaceae and Melastomataceae. With increasing elevation, forest biomass and productivity decrease whereas root-shoot ratio increases (Homeier & Leuschner, 2021).Concerning the belowground perspective, which is the main focus of this work, it is known among these sites that with increasing elevation, average SRL decreases from around 3,000 cm g$^{-1}$ at 1,000 m to 1,500 cm g$^{-1}$ at 3,000 (Pierick *et al.*, 2023). The percentage of root length colonized by arbuscular mycorrhizal fungi (AMF), which interacts with almost all species in the study area (Kottke & Haug, 2004), increases from an average of 25% at the lowest elevation to 61% at the highest (Camenzind *et al.*, 2016). This altitudinal trend, as well as nutrient addition experiments in the sites (Camenzind *et al.*, 2014) suggest that AMF has an important role in plant N uptake in our studied TMF. This is in contrast with the common view that AMF is mostly relevant for P, which has been challenged (Hodge & Storer, 2015).

Principal component patterns for SRL, fine-root diameter, tissue density and N content (Pierick *et al.*, 2021, 2024) are very similar to the global ones described in Bergmann *et al.*, (2020), where fine-root tissue density and N form one axis (i.e. fast-slow gradient) and SRL with fine-root diameter another, in line with the FCG.

### 3.2. Model description

To simulate the trait distributions in the Andes Tropical Montane Forest (TMF), we use the LPJ-GUESS DGVM (Smith *et al.*, 2014) with the NTD (nutrient-trait dynamics) implementation (Dantas de Paula *et al.*, 2021). This model includes individual representations of each tree, a Farquar-based photosynthesis implementation, detailed tree population dynamics (establishment, mortality, disturbances), and abiotic competition processes (light, water, nutrients, space; Smith *et al.*, 2001, 2014). In the NTD version of LPJ-GUESS (see Dantas de Paula *et al.*, 2021) for the full description), plant diversity is included through trait variation, as random values of specific leaf area (SLA) and wood specific gravity (WSG), when new tree saplings are established. These traits are related to further traits, and the trait-trait relationships define major trade-off axes, such as SLA to tissue C:N ratios. For the model in this study, most trait-trait relationships and tissue stoichiometry were not parameterized from global data but from measurements at the study site (Dantas de Paula *et al.*, 2021). The trait combinations each plant receives in establishment impact its competitive success compared to other individuals in any specific environment, with less adapted individuals suffering, e.g., having lower growth than competitors and, therefore, higher growth efficiency mortality (the mortality component in the model related to negative net primary productivity –NPP-values).

Soil organic matter (SOM) dynamics were adopted from the CENTURY model (Parton *et al.*, 1988, 1993, 2010), with organic matter pools and the N and P cycles also included. Organic matter enters the SOM model through vegetation litter input, and total soil organic C for a particular point in time can be estimated by summing up all C in SOM pools. The C:N and C:P ratio of litter, its size (e.g. leaf versus coarse woody debris), soil temperature, soil humidity, and available soil nutrients influence SOM decomposition rates, soil C accumulation and nutrient dynamics. Tissue C:N and C:P ratios also determine nutrient demand, which must be met by root uptake, otherwise photosynthesis and growth become limited by N and/or P. Nutrient limitation also drives increased C allocation to roots, and higher nutrient uptake than demand drives increased C allocation to leaves. LPJ-GUESS-NTD included a simple implementation of AMF-mediated plant nutrient uptake (Dantas de Paula *et al.*, 2021), which was required to reconcile measured available soil nutrient concentrations with observed vegetation structure and trait distributions. This representation was based on Kirschbaum & Paul, (2002), in which vegetation demand for N and P are met by additional uptake from the surface microbial pool. However, this approach had several limitations, among which there was (1) no representation of individual AMF mass per plant, i.e. a "big mushroom" approach (analogous to the "big leaf" approach (Luo *et al.*, 2018)); (2) a fixed AMF colonization rate for all individuals; (3) no C costs for AMF-mediated nutrient acquisition. Also, fine-root traits were fixed among individuals, meaning that the total fine-root surface area for nutrient uptake depended only on fine-root biomass, and fine-root morphological traits such as SRL were not considered. This previous implementation therefore did not provide a realistic relationship between C invested into nutrient acquisition (as fine roots or mycorrhizae) and effective acquisition capacity. Although fine-root biomass in general terms translates into higher nutrient or water uptake capacity, there is high variation for same biomass values, and this is

thought to result from fine-root morphological trait variation (Kokko *et al.*, 1993). Even though fine-root architecture (i.e. root distribution along different soil layers) and rooting depth are considered in our and other models (Langan *et al.*, 2017; Sakschewski *et al.*, 2021), to our knowledge morphological traits such as fine-root diameter and SRL have not been implemented.

### 3.3. Implementation of the fungal collaboration gradient

We consider that fine-root surface area is a better proxy for uptake capacity than fine-root biomass. We consider fine roots as those having a diameter of less than 2 mm, as defined in our reference field samples (Pierick *et al.*, 2021, 2024). The inclusion of SRL as an individual trait permits the calculation of the total fine-root surface area using the equation:

$$A_{root} = SRL \, d_{root} \, \pi \, C_{root}, \tag{1}$$

where $A_{root}$ is the total fine (absorptive) fine-root surface area (m$^2$), SRL is in m/kgC roots, $d_{root}$ is fine-root diameter in m calculated from SRL (see below), and $C_{root}$ is fine-root C biomass (kg C m$^{-2}$). Individuals with higher values of SRL and thinner roots (low diameter) have larger values of $A_{root}$ (since SRL and $d_{root}$ have a non-linear relationship) per fine-root mass. The relationship between SRL and $d_{root}$ was produced using species data measured in our study site (Pierick *et al.*, 2021).

We included SRL as a randomized trait in the tree's individual establishment, in addition to the model´s existing traits SLA and WSG, as well as AMF C mass as a mass pool for each tree. In this updated model version, AMF biomass and area ($C_{AMF}$ and $A_{AMF}$) are implemented as an extension of the root system for each tree individual. For herbaceous vegetation, individuals are not distinguished in LPJ-GUESS. Accordingly, $C_{AMF}$ and $A_{AMF}$ are not individual-based for herbaceous vegetation, with only one pool for $C_3$ and one pool for $C_4$ herbaceous vegetation. The total hyphal surface area for each individual's AMF is calculated as in equation (1), substituting $A_{root}$, $d_{root}$ and $C_{root}$ for $A_{AMF}$, $d_{AMF}$ and $C_{AMF,}$ respectively, and SRL for specific hyphal length (SHL). Values for SHL and $d_{AMF}$ (Table A1) are fixed for all individuals (no fungal trait variation) and based on Raven *et al.*, (2004). Due to their very thin hyphae, AMF typically has specific surface areas that are several times larger than those of fine roots with the same C mass Raven *et al.*, (2004). The total soil area a plant individual can explore for nutrients is thus equivalent to $A_{root} + A_{AMF}$. Cooperation with AMF benefits a plant individual, increasing the surface area for its nutrient uptake to levels which cannot be reached by its fine roots alone. However, cooperation with AMF implies in fungal $C_{AMF}$ costs which must be borne by the plant. We expect then that in more nutrient limited environments, such as in higher elevation areas, mycorrhiza colonization will be high. On the other hand, in lower elevation sites where light competition plays a large role, investment in mycorrhiza will not lead to higher fitness, leading to lower colonization rates.

AMF C dynamics for each individual's $C_{AMF}$ were implemented similarly to the structural plant C compartments, i.e., leaf, wood, and fine roots. AMF respiration is included and depends on the current daily $C_{AMF}$ and a fixed AMF C:N ratio value (Orwin *et al.*, 2011). C is allocated to AMF biomass ($C_{AMF}$) at each timestep , and calculated as:

$$\Delta C_{AMF} = \Delta C_{inc} \cdot r_{max} \cdot m_{col} , \tag{2}$$

where $\Delta C_{inc}$ is the total biomass increment for this individual in this timestep (NPP – respiration) and $r_{max}$ is a calibrated parameter defining the maximum fraction of $\Delta C_{inc}$ (0 - 1) transferred to AMF for growth. The individual's mycorrhizal colonization rate $m_{col}$ $(0 - 1)$ was defined from a correlation from $d_{root}$ based on global trait data from the GROOT database (Guerrero-Ramírez *et al.*, 2020), since our study area had no species-level (only site-level) measurements of mycorrhizal colonization rates. Based on the global $m_{col} - d_{root}$ relationship, we assume that increasing colonization rates will result in a larger C transfer from plant to AMF, in other words, increased C costs. The parameter $r_{max}$ is therefore calibrated to establish the incline of the assumed linear relationship between AMF colonization and C increment. We explore its values from 0-1 to determine values in which maximum benefit for the plant occurs. A graphic visualization of the relationship between $m_{col}$ and $r_{max}$ can be seen in Fig. A1.

In the new model, ecological filtering determines which AMF colonization rates are optimal, leading to the highest biomass growth rates under given environmental conditions. Those tree individuals with the optimal AMF colonization then outcompete other individuals. Competition occurs for light, water, N and P. This approach to C transfer between plants and AMF differs from those in which N and P are exchanged for fixed or varying costs of C (Fisher *et al.*, 2010; Allen *et al.*, 2020; Reichert *et al.*, 2023) in favor of a more explicit representation of AMF physiology. Although fungal C demand is represented here, we did not include N or P demand and uptake of the mycorrhiza themselves. Thus, all nutrients absorbed by the AMF hyphae in the model are transferred to the plant (Fig. 1).

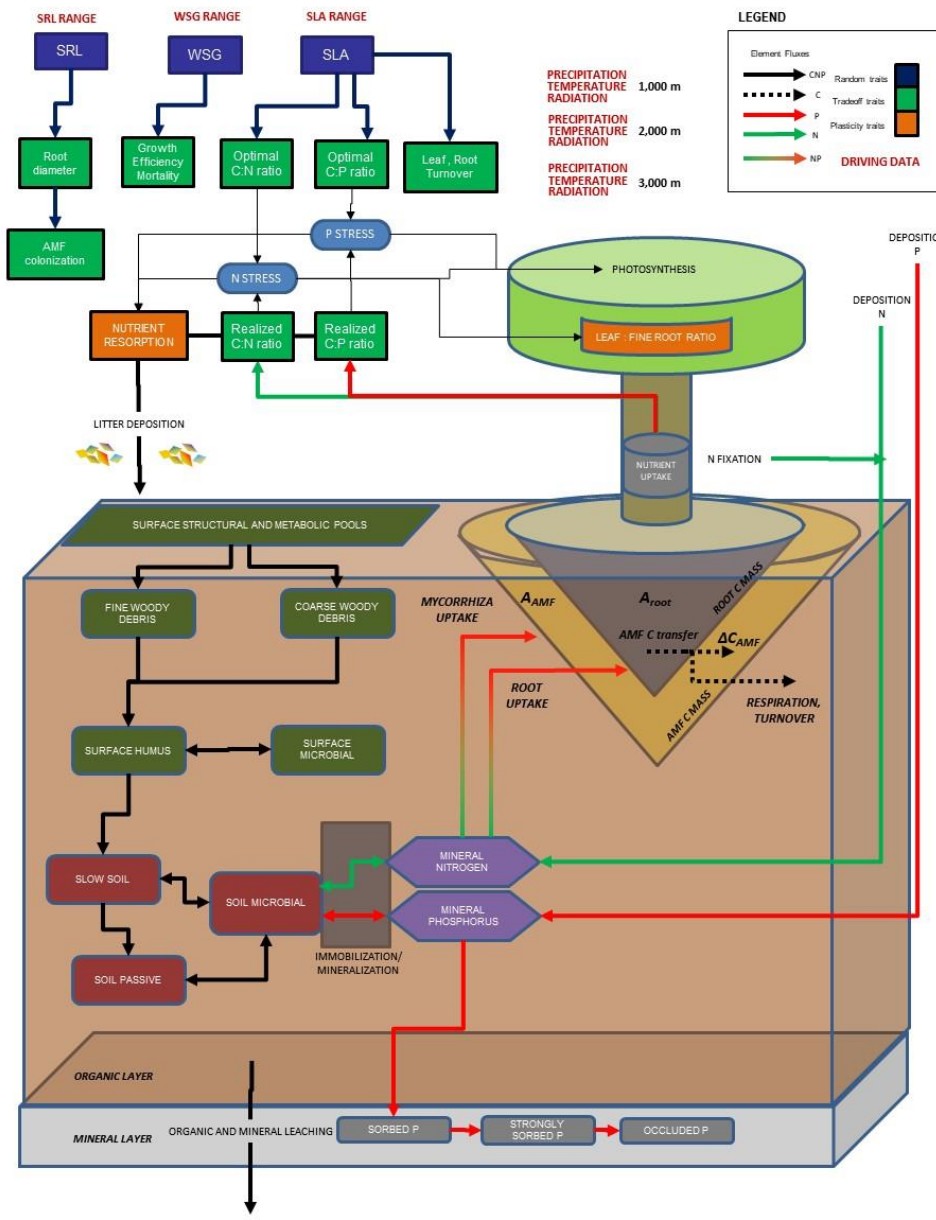

**Figure 1. Full Schematic of LPJ-GUESS-NTD including the new fine root and mycorrhiza implementations in relation to** (Dantas de Paula *et al.*, 2021)**. For more details, refer to that publication. In the present study, Arbuscular Mycorrhiza Fungi (AMF) was added as an individual component of a plant. A$_{root}$ and its analog A$_{AMF}$, as described by equation (1) and $\Delta C_{AMF}$ from equation (2)**

are indicated. Dark green and red elements represent soil organic matter (SOM) pools, while the purple hexagons represent mineral (plant available) pools.

Relevant to our model approach for the FCG implementation is the advantage posed by the AMF's higher nutrient
absorption area and uptake capacity. Literature research has yielded little data on the necessary Michaelis-Menten equation parameters for model runs: the maximum uptake rate of N and P (Vmax) and the half-saturation constant (Km) (Silveira & Cardoso, 2004; Pérez-Tienda *et al.*, 2012). These values represent respectively the amount of nutrients taken up per fine-root mass for each time step and how well the uptake occurs under low concentrations. Values for AMF indicated that due to different N and P form transport systems, fungi differ in their uptake dynamics to fine roots (Silveira & Cardoso, 2004; Wu
*et al.*, 2020). Particularly in the Km values, as AMF can absorb N and P under much lower concentrations than fine roots (Table A1).

Another relevant parameter in the simulations was the turnover rate of AMF extra radical mycelium. This has been recognized as a complex field measurement, with estimations as low as five to six days (Staddon *et al.*, 2003) and nine days (Godbold *et al.*, 2006). However, here we consider the extra radical mycelium longevity of 80 days as proposed by Raven *et*
*al.*, (2018), based on the elongation rate of up to 3 mm d$^{-1}$ for AMF hyphae (Olsson & Johnson, 2005; Smith & Read, 2010). Graphs of the data used in the trade-offs and the equations used in the trait-trait relationships, can be seen in Fig. S2.

### 3.4. Model drivers and evaluation data

Each of our three elevation sites was simulated with distinct daily data of temperature, precipitation, and radiation,
interpolated from measured monthly average values from climate station data at each elevation site between 1999 and 2018 (Peters & Richter, 2009; Rollenbeck *et al.*, 2015; Bendix, 2020). Soil data and parameters were taken from the World Soil database (FAO/IIASA/ISRIC/ISS-CAS/JRC, 2012); N and P deposition rates used were from Dantas de Paula *et al.*, (2021); which were measured weekly during the same 1999 – 2018 period. Details on how driver data influences ecosystem processes can be found at Smith *et al.*, (2014) and Dantas de Paula *et al.*, (2021).
Minimum and maximum SLA, WSG and SRL values for initialization of tree individuals at establishment were defined using measured values from the field (Báez & Homeier, 2018; Homeier *et al.*, 2021; Pierick *et al.*, 2023). These minimum and maximum observed trait values consider all three elevation sites as one since we do not include dispersal or establishment limitation along this elevational range. Regarding the trade-off relationship of the traits SLA, WSG, and SRL, we use for the first two the same correlations defined in (Dantas de Paula *et al.*, 2021). For SRL, we derive $d_{root}$ from it,
using the dataset of Pierick *et al.*, (2023), and AMF colonization rates from $d_{root}$ and $m_{col}$ values from the GRooT database (Guerrero-Ramírez *et al.*, 2020). Equations, $R^2$ values, and graphs can be seen in Fig. 2. Model parameterizations can be found in Table A1.

### 3.5. Modeling protocol and scenarios

Following the standard LPJ-GUESS procedure (Smith *et al.*, 2014), the LPJ-GUESS-NTD model was initialized from the bare ground and allowed to spin up for 500 years using the driving data to build vegetation and soil C, N and P stocks. Following the spin-up period, a further 200 years was simulated for generating the modelled results by repeating the observed climate data to represent long-term mean conditions. We repeated this process for two defined scenarios: (1) no AMF collaboration (AMF-off), in which the parameter for maximum C transfer to AMF $r_{max}$ is set to zero, and $A_{total}$ equals

$A_{root}$; (2) AMF collaboration (AMF-on), considering the new implementations defined in the current study and where uptake of nutrients depends on $A_{root} + A_{amf}$ (Fig. 2). Each simulation was set to represent a total area of 10 hectares, and the whole procedure was replicated 30 times to average random effects.

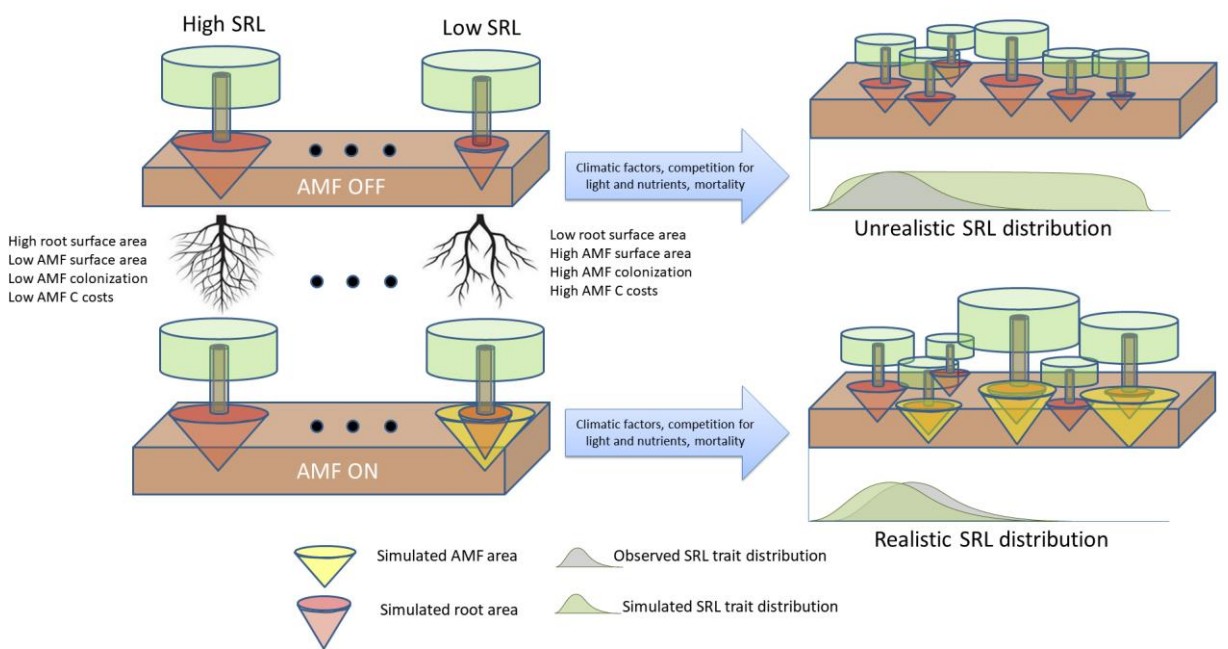

**Figure 2. Scenarios used in this study to evaluate the role of the fungal collaboration gradient in shaping Specific Root Length (SRL) distributions. In the top scenario (AMF-off), transfer of C to Arbuscular Mycorrhiza Fungi (AMF) is deactivated ($r_{max} = 0$) and plants depend only on roots to acquire nutrients. In the scenario below (AMF-on) transfer of C is active ($r_{max} = 0.5$) and AMF participates in nutrient acquisition.**

### 3.6. Model evaluation

To assess model results, we compared them to the trait and forest structure measurements also used in Dantas de Paula *et al.*, (2021), in addition to SRL, fine-root diameter, and AMF colonization data reported from Pierick *et al.*, (2021) and Camenzind *et al.*, (2016). The trait distributions from these same sources as well as mean values for SLA, WSG, SRL, and fine-root AMF colonization, were also used to evaluate the simulated trait distribution results for each of the three elevation sites.

### 3.7. Sensitivity Analysis to infer maximum carbon fraction transferred to AMF

We included a sensitivity analysis of the parameter $r_{max}$ (the fraction of plant NPP allocated to its growth in each timestep transferred to AMF when colonization rate is 100%, or a rate of 1), which is the only parameter that is calibrated and not based on observations. The sensitivity analysis allowed us to assess the full range of possible $r_{max}$ values (0 - 1), and identify optima of this parameter which maximizes plant productivity and plant-fungi interactions. Given the lack of experimental data and relationships, we assumed here a linear response between AMF colonization rate and $r_{max}$ (see equation (2)).

We executed 30 simulations for each elevation site with the $r_{max}$ parameter varying from 0 to 1 and examined the effects on traits and ecosystem processes, as well as the total rate of transfer of C from plant to fungi, and C cost per N or P uptake.

## 4. Results

### 4.1. Sensitivity analysis of the $r_{max}$ parameter

With increasing maximum C transfer rate from plant to AMF ($r_{max}$), the model simulated an increase in average biomass and productivity for all sites (Figs. 3a-b), and an increasing community average AMF colonization rate (Fig. 3c), reflecting the alleviation of N and P limitation (Figs. A3e-f) and increasing benefit for plants in interacting with fungi. This pattern continued until an $r_{max}$ of 0.25-0.5, when productivity and biomass reach their highest values. At that point, average community AMF colonization reach a maximum, having values which are closest to observations. With increasing rmax, average colonization rates decrease, indicating a decoupling of plant and fungi. For this reason, we picked a value of 0.5 $r_{max}$ for subsequent analysis. We consider the simulations with a $r_{max}$ of 0.5 as the value where the largest benefit for the plant, when interaction with AMF occurs, and consider the AMF-on scenario to be $r_{max} = 0.5$. More details and results for the sensitivity analysis of $r_{max}$ can be seen in the appendix A.

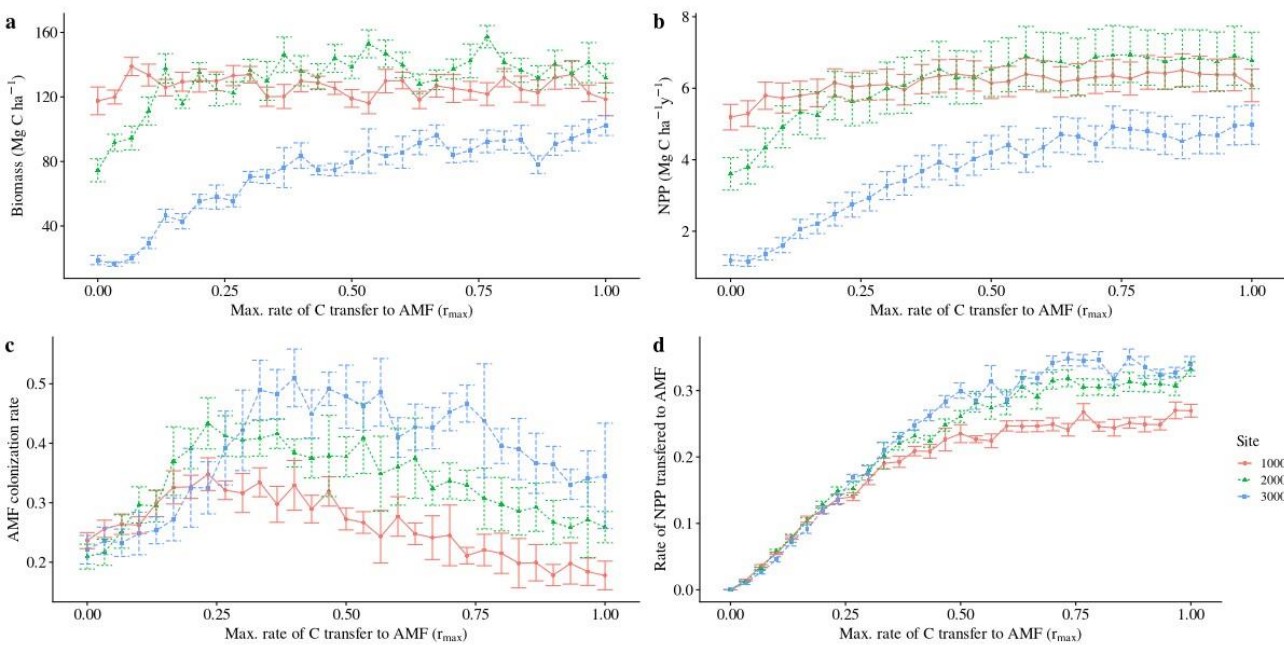

**Figure 3. Sensitivity analysis of the parameter $r_{max}$, maximum C allocation to growth for mycorrhiza. Each point represents 200-year averages using a particular $r_{max}$. Whiskers indicate +/- SD from the 200 years of each run. Red lines: 1,000 m. Green lines: 2,000 m. Blue lines: 3,000 m.**

### 4.2. Influence of the fungal collaboration gradient on plant traits

The decrease in SRL and increase in AMF colonization rates observed in the field data along the three elevation gradients were reproduced by the model in the AMF-on scenario. Simulated SRL and AMF colonization rates were within the confidence interval of field measurements for the 1,000 and 2,000 meter a.s.l. sites (Fig. 4a-b), ranging from 4,929.31 – 3,339.07 cm g-1 (SRL) and 0.29 – 48 (AMF col.) as can be seen in Table 1. When AMF was deactivated, no differences between elevations were found for simulated SRL and AMF colonization (Fig. 4a, b).

The distribution graphs of SRL showed clearly how the fine-root traits are shaped by the relationship with AMF for this environment (Fig. 5a). When the FCG was active (AMF-on), the SRL distribution was much closer to the observed distribution than AMF-off (Fig. 5a). The resulting AMF-off SRL pattern was very similar to the initial uniform random distribution at the establishment, with a lack of trends and no differences in SRL between the three elevational sites. Viewing SLA in this form as well, it can be seen that the AMF-off community was significantly more conservative (ranging from 350 47.96 – 75.18 cm² g$^{-1}$) than AMF-on (ranging from 88.79 – 104.97 cm² g$^{-1}$) (Table 1). A similar pattern was found regarding leaf stoichiometry, as C:N and C:P ratios reflected SLA values and also became more conservative (larger values, with

higher proportion of C) when AMF was deactivated (Table 1). WSG, on the other hand, was unaffected by the different AMF scenarios. Model performance thus improved in the AMF-on scenario, reducing median absolute errors for SRL in comparison with the AMF-off. SLA on the other hand was slightly worse with the activation of AMF uptake (Table A2, Figure A5).

| | Units | 1000 | | | | 2000 | | | | 3000 | | | |
|---|---|---|---|---|---|---|---|---|---|---|---|---|---|
| | | AMF-on | | AMF-off | | AMF-on | | AMF-off | | AMF-on | | AMF-off | |
| | | mean | SD | mean | SD | mean | SD | mean | SD | mean | SD | mean | SD |
| Vegetation Biomass | Mg ha$^{-1}$ | 126.19 | 5.07 | 119.33 | 6.46 | 135.92 | 7.51 | 83.99 | 5.80 | 84.90 | 4.85 | 16.49 | 2.62 |
| NPP | Mg ha$^{-1}$ y$^{-1}$ | 6.30 | 0.10 | 5.21 | 0.12 | 6.51 | 0.17 | 3.75 | 0.10 | 4.28 | 0.09 | 1.16 | 0.06 |
| Soil C stocks | Mg ha$^{-1}$ | 29.53 | 0.42 | 24.08 | 0.56 | 43.53 | 0.75 | 24.91 | 0.60 | 49.57 | 1.26 | 15.65 | 0.55 |
| net N mineralization | kg ha$^{-1}$ y$^{-1}$ | 37.80 | 0.86 | 41.46 | 0.63 | 39.72 | 0.90 | 29.03 | 0.73 | 22.23 | 0.71 | 10.58 | 0.28 |
| SLA | cm² g$^{-1}$ | 104.97 | 3.69 | 75.18 | 2.54 | 100.12 | 3.53 | 59.17 | 1.90 | 88.79 | 5.05 | 47.96 | 1.07 |
| SRL | cm g$^{-1}$ | 4929.31 | 195.32 | 5975.69 | 156.66 | 3996.16 | 182.40 | 5885.44 | 165.88 | 3339.07 | 194.90 | 5943.29 | 105.32 |
| AMF colonization | - | 0.29 | 0.02 | - | - | 0.39 | 0.02 | - | - | 0.48 | 0.02 | - | - |
| C:N leaf | - | 39.56 | 1.01 | 46.45 | 0.73 | 42.34 | 1.09 | 51.49 | 0.70 | 44.49 | 1.05 | 52.31 | 0.78 |
| C:P leaf | - | 852.30 | 26.16 | 1001.23 | 18.52 | 974.52 | 30.98 | 1256.29 | 21.02 | 1069.53 | 32.24 | 1342.42 | 22.99 |
| C:N litter | - | 32.87 | 0.67 | 41.63 | 0.61 | 35.16 | 0.99 | 46.85 | 0.64 | 37.39 | 0.84 | 47.85 | 0.74 |
| C:P litter | - | 678.48 | 16.47 | 877.86 | 14.87 | 776.20 | 26.75 | 1136.88 | 19.22 | 875.47 | 25.88 | 1245.15 | 20.93 |

Table 1. Simulated results for the two AMF treatment scenario AMF-on (0.5 $r_{max}$) and AMF-off (0.0 rmax) along the three elevation sites (1,000 m.a.s.l., 2,000 m.a.s.l., 3,000 m.a.s.l.). T-tests (N = 30) between AMF-on and AMF-off scenarios presented significant differences for all variables, as well as t-tests between elevations (1,000 compared to 2,000 and 1,000 compared to 3,000), except for SRL, which between elevations did not present significant differences.

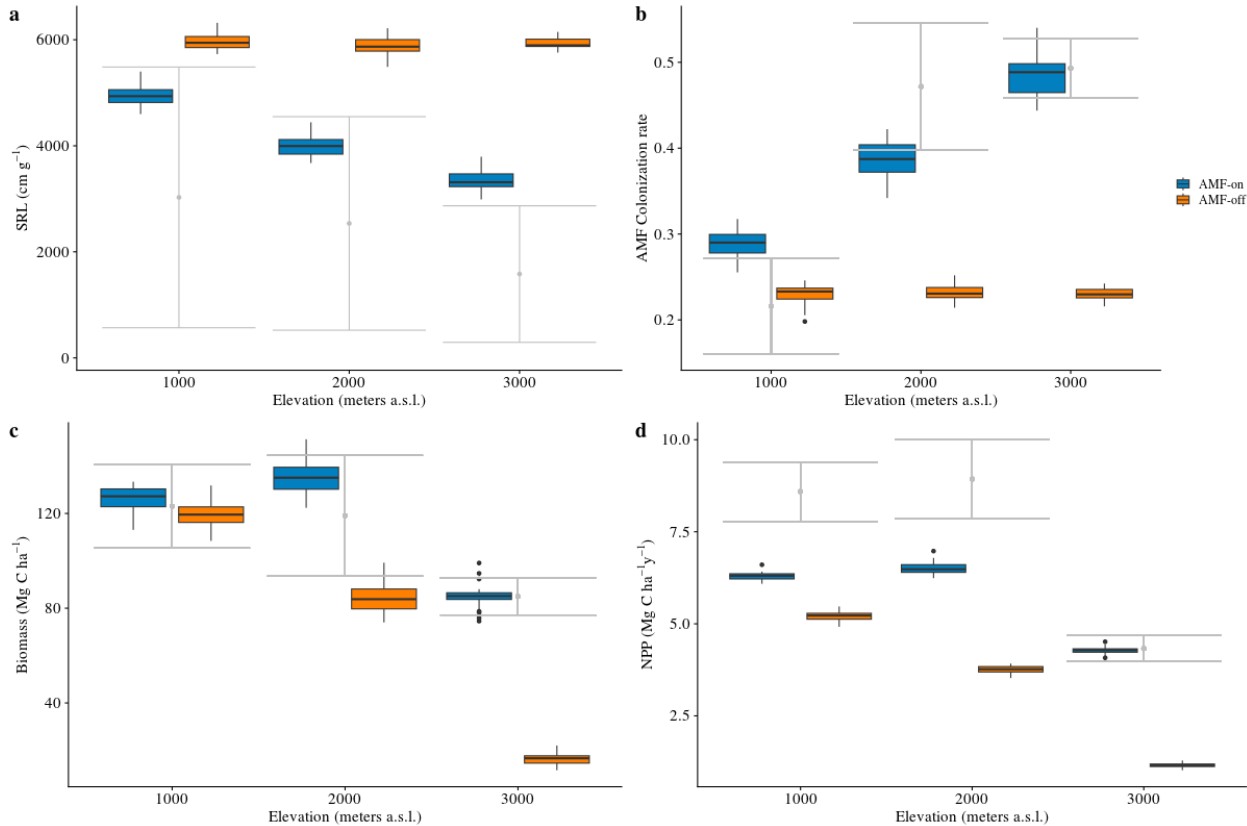

**Figure 4. Results of simulated and observed C-related values for the elevation gradient, (a) specific root length – SRL, (b) AMF colonization rate, (c) biomass (sum of above and belowground) and (d) NPP – annual net primary production,. Scenarios are AMF-on – C allocation to growth $r_{max}$ = 0.5; AMF-off – C allocation to growth $r_{max}$ = 0.0. Grey bars are field measurements, with whiskers indicating confidence intervals.**

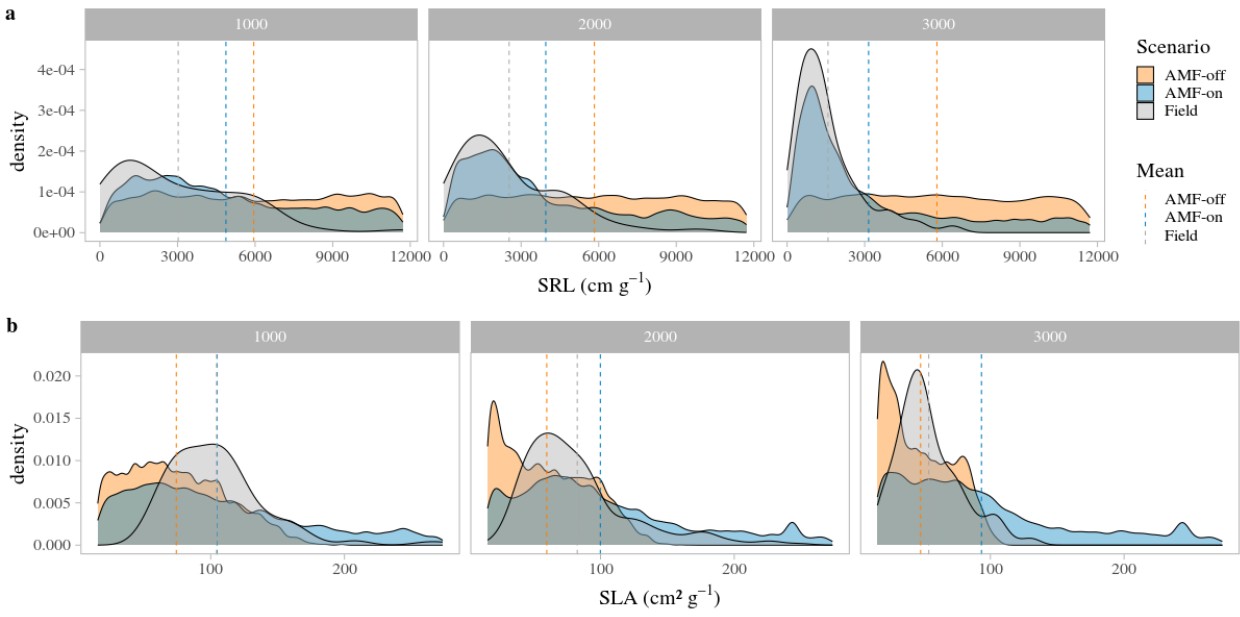


**Figure 5. Trait density distributions for (a) specific root length - SRL and (b) specific leaf area - SLA: Scenarios are AMF-on – C allocation to growth $r_{max}$ = 0.5; AMF-off – C allocation to growth $r_{max}$ = 0.0. Vertical dashed lines are mean values for each scenario.**

### 4.3. Influence of the fungal collaboration gradient on carbon stocks and productivity

Simulation results with activated AMF (AMF-on scenario, $r_{max}$ = 0.5) exhibited on average higher biomass and productivity values than AMF-off. Simulated biomass in AMF-on simulations were all within observed bounds whereas with AMF-off, only the 1000m site produced biomass within observed bounds (Fig. 4). Biomass differences between the AMF-on and AMF-off scenarios were the largest for the 3,000 m elevation site, where the former had 84.9 Mg C ha$^{-1}$ +/- 4.85 SD and the latter 16.49 Mg C ha$^{-1}$ +/- 2.62 SD, an 80.6% reduction (Fig. 4c, Table 1). Similar differences were found regarding NPP between both scenarios and when compared to observations, AMF-on yielded better NPP estimates, in particular for the 3,000 m site (Fig. 4d).

Average soil C stocks increased in the AMF-on scenario from 29.53 at 1,000 m to 49.57 Mg ha-1 at 3,000 m, while in the AMF-off scenario it decreased from 24.08 to 15.65 Mg ha$^{-1}$ (Table 1). Although Net N mineralization rates where slightly higher in the AMF-off (41.46 kg ha$^{-1}$ y$^{-1}$) in comparison with the AMF-on (37.80 kg ha$^{-1}$ y$^{-1}$) in the 1,000 site, it decreased to more than half in the 3,000 site, from 22.23 to 10.58 kg ha$^{-1}$ y$^{-1}$ (Table 1). Litter stoichiometric properties reflected leaf C:N and C:P ratios, becoming more carbon rich (Table 1).

Total AMF biomass peaked at the 2,000 m elevation site with around 0.45 Mg C ha$^{-1}$, and was lowest at the 1,000 m elevation site with 0.35 Mg C ha$^{-1}$ (Fig. A3b). However in relative terms to total vegetation biomass, AMF increased with elevation, reaching around 0.4% of total plant biomass stocks at the highest elevation site (Fig A4).

## 5. Discussion

Trait-based dynamic vegetation models (DVM) represent an important tool which can be used to test theoretical frameworks of functional diversity. Our simulated results suggest that the fungal collaboration gradient (FCG) is a major driver of fine-root functional diversity and ecosystem processes. We were able to compare three contrasting environments across altitudes, and the simulations largely followed field observations in magnitude and trend, but only when the FCG was activated in the model. These results have important implications for understanding the role of fine-root trait syndromes in driving ecosystem processes, as well as for conservation strategies and ecosystem management.

### 5.1. The importance and implications of the fungal collaboration gradient for fine-root trait diversity and ecosystem processes

Our results show that the FCG is a relevant concept for linking fine-root trait diversity and ecosystem functioning. The FCG allowed realistic fine-root trait distributions to emerge from the simulations. The FCG also has an important role in vegetation productivity and the organic C cycle. The lower productivity and biomass of AMF-off communities suggest that the lack of mycorrhiza is detrimental to plant nutrition and productivity at the higher elevations in our study regions. The deactivation of C transfer from plant to mycorrhiza represents a long-term fungal suppression experiment from an empirical ecology perspective. These are incredibly valuable in establishing the causality of interactions, but they are understandably difficult to perform under natural conditions in long-standing ecosystems. Field studies which tested either the removal of mycorrhiza or their inoculation show responses proportional on their relevance for the dominant plant species in an ecosystem (Lin *et al.*, 2015). When mycorrhizas are relevant for a community, we can expect that fungal suppression will significantly reduce total biomass, as for example, in O'Connor *et al.*, (2002). In that study, a heavily AMF-colonized herb species had a reduction of 60% of biomass after the application of fungicide, which is comparable with the 80% reduction in our highest elevation site. A meta-analysis of AMF experimental inoculation effects to plant growth found that on average AMF increase biomass in 47% (Wu *et al.*, 2024). Some small-scale field studies that suppressed fungal colonization in intraspecifically diverse plant populations through molecular, rather than chemical, methods are perhaps able to more cleanly determine the role of AMF on biomass, and have supported altered biomass and competitive ability of plants in the absence of the ability to associate with AMF (McGale *et al.*, 2020; Groten *et al.*, 2023).

The presence of AMF at the higher elevation sites was not enough to drive the biomass in the 3,000 m sites to the same values found at 1,000 m, as observed. As evaluated in a climatic sensitivity analysis in a previous LPJ-GUESS-NTD

publication, the gradient of biomass and productivity in our study sites is driven ultimately by temperature impacts on nutrient cycling (Dantas de Paula *et al.*, 2021). In the model, lower decomposition rates result is less nutrient mineralization and availability, which is exacerbated by changes in leaf and litter traits (more C in relation to N and P, tougher leaves which decompose poorly), and limit productivity and biomass. Field studies complement this, indicating that with elevation more biomass is allocated belowground suggesting higher competition for nutrients (Leuschner *et al.*, 2013).

Related to this impact on plant growth are the effects of AMF to soil C stocks and nutrient dynamics. This is a complex topic since in the literature the presence of AMF has been related to reduction (Wurzburger & Brookshire, 2017) or increase (Rillig *et al.*, 2001) of soil organic C. Our modelling approach can provide insights to this contrasting effect – AMF may both increase plant productivity (by alleviating nutrient limitation) which would tend to increase soil C stocks, while at the same time driving more labile litter and accelerating decomposition rates which would reduce soil C. Since LPJ-GUESS-

NTD includes both litter deposition and stoichiometry (which is a direct consequence of plant trait variation), the balance between increase in soil C input and higher soil C turnover can be effectively disentangled depending on the simulated site. For the particular area simulated in this study, our model suggests that AMF drives higher soil C stocks due to its support of a higher plant productivity (Table 1), particularly at the highest elevation site. This occurs in spite of AMF-off having lower net N mineralization rates and higher C:N and C:P litter (Table 1), which promotes lower decomposition and thus soil C

accumulation. Exploring if the model could reproduce patterns of lower soil C stocks with AMF presence in other regions would be both an important model performance test and help to estimate how global average soil C stocks are affected by AMF. Therefore, the value and necessity of a DVM, as implemented here, are evident in expanding our understanding of these results across diverse natural ecosystems and sites. Moreover, this approach is crucial at a scale that is urgently needed in the context of climate change, particularly for conservation efforts.

Our modelling approach also reveals linkages between belowground and aboveground traits. For our simulations, we observed that more nutrient-limited environments had both low specific leaf area (SLA) and low SRL (high diameter). In addition, deactivating AMF (AMF-off) resulted in communities with significantly lower SLA values. In other words, constraining fine-root diversity (i.e. by excluding AMF) in our model resulted in aboveground changes in leaf traits and a reduction in aboveground trait diversity (SLA range). These results reveal that belowground traits and the FCG can

significantly influence aboveground vegetation traits across different altitudes in a tropical system. This is new information for this system and is rarely explored in trait research or DVMs, likely because the complex interactions are hard to untangle in field experiments or observational studies.

## 5.2. Model advancements and perspectives on plant-fungal interactions

We have presented a relatively simple and effective implementation of the root collaboration gradient, which can be included in other DVMs. A crucial step in the model development was to translate fine root and hyphal mass into the total surface area through specific root and hyphal length (Eq. 1). Implementing the fine-root surface area calculation based on a varying length-to-mass variable within observed ranges allowed the modelled fine roots to function using observational fine-root physiological parameters for uptake kinetics (Table A1). Fine-root surface area, in particular at the highest elevation site, is

clearly too low to support nutrient uptake and vegetation growth without mycorrhiza. As the model implementation of fine-root nutrient absorption was changed from based solely on fine-root mass to fine-root surface area, it became clear that mycorrhizal associations (using observed physiological parameters) were necessary to support observed vegetation biomass under observed soil nutrient concentrations and nutrient demands, especially for the highest elevation site in our study.

Our approach for linking plant C expenditure to nutrient acquisition differs from other approaches since we (1) explicitly

consider fungal C mass and turnover in the model, allowing for its future measurement and validation in the field; (2) plant C investment into mycorrhiza is divided into fungal growth and tissue respiration and (3) costs for nutrient uptake are not predetermined in the model but dynamic, varying not only between plant individuals due to fine-root traits (i.e. AMF colonization rates) but also within an individual`s life history since C must be allocated to sustain fungal respiration (dependent on fungal C mass which varies with time). In addition, we consider individual plants in our model, which

compete for nutrients that result from a dynamic organic matter submodel, and consider feedbacks from plant tissue stoichiometry. Finally, LPJ-GUESS-NTD is a trait-based DVM and not PFT-based (Plant Functional Type, where simulated individuals from the same category have the same traits), allowing the exploration of plant functional diversity.

Modelled C costs per N or P uptake (Fig. A3) increase with elevation, as expected due to the lower nutrient content of the soils in the highest sites. These costs are model-emergent, and are outcomes of C investment processes in AMF tissue

growth and respiration and nutrient acquisition through uptake depending on plant and fungi structure and physiology. The approach used in our study is thus different from the ones followed by other models (Allen *et al.*, 2020; Reichert *et al.*, 2023), where `C costs of nutrient uptake or their ranges are defined a priori. The costs defined here are thus connected to other processes within the model, and reflect field-measured parameters for AMF dynamics and structure as defined in Table A1.


## 5.3. Model limitations and links to observations

Gaps in the current understanding of plant-fungal interactions present a challenge in modeling their processes (Makarov, 2019; Hawkins *et al.*, 2023), and limit the projection of their effect on larger-scale processes such as the C cycle. Key processes which are poorly constrained by data are fine root and mycorrhiza uptake of nutrients, C transfer between plant

and fungi, their tissue turnover rates and longevity, mycorrhiza C:N ratios (impacts mycorrhiza respiration and total C costs)

and the N and P demand of mycorrhiza tissue. In addition to parametrization, field data is also invaluable for model evaluation. With regards to this study, our modeled total AMF mass per hectare reached values of up to 0.45 Mg C ha$^{-1}$, representing around 0.5% of total plant biomass, assuming an extraradical mycelium (ERM) longevity of 80 days. Godbold *et al.*, (2006) estimated higher values of around 1.1 Mg C ha$^{-1}$ for the EuroFACE sites, and an ERM longevity of 9 days.

These values may however be at an upper range of observations, as Parihar *et al.*, (2020) found AMF biomass values between 0.054 and 0.9 Mg C ha$^{-1}$, which supports our choice of AMF turnover parameter in Table A1. In any case, ERM longevity and total fungal biomass are invariably linked, meaning that increases in the former will result in increases in the latter. Therefore, more measurements of both would be invaluable for models to parameterize these two important components of AMF dynamics. Such data, while in many cases difficult and costly to measure, could be very important for

future simulations and foster collaborations between modellers and experimental and field ecologists. For example, by exploring the parameter ranges of turnover (as we have done with C investment, varying the $r_{max}$ parameter), models can estimate which AMF turnover values produce viable plant communities, which then can be tested against field estimations. This would be particularly important for large spatial and temporal scale estimations of increasing atmospheric CO2 effects on soil C stocks, since AMF turnover rates influence soil C (Treseder & Allen, 2000).

The new empirical relationships between SRL, fine-root diameter and AMF colonization included in the LPJ-GUESS-NTD model for this study have led to satisfactory results. However, the relationship between AMF colonization rate and fine-root diameter as estimated in the field, showed considerable variation (Fig. 2). This indicates that although the relationship between fine-root diameter and AMF colonization is considerable, there is much room for improved understanding on this topic. More thorough studies involving fine-root anatomy, physiology and traits may uncover further relationships and

variation axes which would be invaluable to help improve vegetation models.

How realistic is our modelled relationship between AMF colonization and plant C transfer? In the lack of field-based empirical evidence, particularly for the type of environment we simulate, our modelling study represents a theoretical estimate of the link between mycorrhizal colonization rates and C transfer from plant to AMF as a fraction of NPP. The sensitivity analysis suggests a general agreement of equation (2) to observations with an $r_{max} = 0.5$. The value used for our

simulations of $r_{max} = 0.5$ resulted in around 30% of NPP (Fig. 3d) or 20% of GPP allocated to AMF (including AMF respiration, Fig. A3g). This value is higher than the average values suggested in the literature (Chapin III *et al.*, 2011). Řezáčová *et al.*, (2017) suggested that the average GPP expenditure of plants to AMF is less than 10%, implying less than 20% of NPP. However, most results exist for crops. Wild plants such as those in our study, can allocate substantially more C to AMF (Hawkins *et al.*, 2023). We conclude that the high $r_{max}$ values in our results, might be realistic under severe nutrient

limitation and fine-root AMF colonization of around 50%, as observed in our highest elevation site (Pierick *et al.*, 2023). More field research of this maximum transfer rate (having e.g. measurements of NPP, AMF C transfer and fine-root intra- and extraradical colonization) would be invaluable to better constrain this important component of NPP, as well as a confirmation of whether the relationship between C transfer and colonization rate is in fact linear, as assumed here. A better

understanding of this link would be invaluable for management practices to improve estimations of plot-level plant to soil C
transfer and AMF biomass.

### 5.4. Future directions in model development, expansion of our approach and model application

Our exploration of fine-root traits and the FCG was limited to three sites in the tropical mountain forests of Ecuador. These areas were chosen due to data availability, and are limited in spatial scope, but we argue that our results are relevant in
general since (1) the axis of trait variation observed for our sites (Pierick *et al.*, 2021, 2023) followed closely by those estimated globally (Bergmann *et al.*, 2020), and (2) AMF is by far the predominant mycorrhizal type both phylogenetically (70% of plant species), in land cover (55%), and in the global NPP contribution from their plant hosts (63%; Hawkins et al. 2023).

Other nutrient acquisition strategies may be more relevant regionally. Although in our studied sites foraging strategies
involving AMF are prevalent in nutrient-poor areas, in strongly P-limited areas of the Amazon lowlands nutrient acquisition strategies such as the exudation of phosphatases or organic acids might also play a significant role (Reichert *et al.*, 2023). The implementation of exudation C-investment strategies may be necessary to account for increased fitness in nutrient poor environments and allocation of C belowground, as up to 16% of NPP was found to be allocated to the production of organic acids in a P-limited environment (Aoki *et al.*, 2012).

The inclusion of other mycorrhizal types could also be important for correctly simulating certain environments. The second most important mycorrhizal type, ectomycorrhizal fungi (EMF), is dominant in the boreal and some tropical regions and dominates in 25% of the Earth's terrestrial ecosystems (Hawkins *et al.*, 2023). EMFs can receive on average higher fractions of plant productivity as C transfers (Hawkins *et al.*, 2023), have different pathways for nutrient uptake (Phillips *et al.*, 2013), and may have a distinct effect on the soil and vegetation C dynamics (Terrer *et al.*, 2021). Since EMF can actively extract
nutrients from non-labile sources, they may allow higher plant productivity than AMF for environments where labile sources are extremely poor, and affect simulated soil C stocks and fluxes. For instance, one modelling study showed how EMF presence can affect C storage (Moore *et al.*, 2015). Regarding the FCG, EMF are expected to follow the SRL trends as AMF (Bergmann *et al.*, 2020), meaning that the main difference for future EMF model implementations might be to account for organic nutrient acquisition kinetics. The inclusion of EMF into DVMs should consider important differences to a AMF
implementation. First, the access EMF has to organic sources requires the production of specialized enzymes, which are not present in AMF. This means that total cost borne by plants for nutrient acquisition may be higher in EMF symbiosis, as confirmed empirically (Hawkins *et al.*, 2023). Second, kinetic uptake parameters, fungal tissue C:N ratios (which in our model impacts mycorrhiza respiration), turnover rates as well as SHL traits should differ due to distinct EMF physiology and morphology. Within the trait-varying approach, one possible implementation could be randomizing the individual's
preference for AMF or EMF (or no mycorrhiza) during establishment (but not for grasslands, which are AMF exclusive). This can be an interesting prospect to test whether AMF or EMF dominance could emerge in a global model following

climatic and edaphic conditions, where AMF would dominate in areas with predominantly mineral soils and EMF where soil organic matter has higher stocks or fluxes (Read, 1991). Also, if AMF and EMF effects on soil C stocks differ under increasing $CO_2$, as been suggested (Terrer *et al.*, 2021), and if mycorrhiza are indeed the primary controls of the CO2

fertilization effect (Terrer *et al.*, 2016).

Large-scale simulations and future projections are expected next steps for trait-based DVMs, such as the one included in this study. Global or regional maps of key vegetation traits have recently been published using several methods, such as combining machine learning and remote sensing, and represent a new way of showcasing worldwide variation in plant attributes (Moreno-Martínez *et al.*, 2018). These maps can be used in turn to evaluate similar maps from trait-based DVMs,

which can provide further ecological information that cannot be sensed from satellites, i.e., on belowground traits and processes. Regarding future projections, an obvious follow-up study to this one would be changes in fine-root traits and fungal collaboration under climate change scenarios. An analysis of data from Free Air CO2 Enhancement (FACE) experiments showed that the mycorrhiza type may determine the extent of CO2 fertilization effects (Terrer *et al.*, 2018, 2021), which is a significant uncertainty in future projections of vegetation and organic C cycle changes (Hickler *et al.*,

2015; Walker *et al.*, 2021). More explicit treatments of mycorrhiza in DGVMs and land surface models might reduce this uncertainty.

Finally, our study can provide important insights into the recent discussion on the importance of AMF for ecosystem management, in spite of its limitations. First, the relevance of AMF abundance and diversity for agriculture has been argued for (Rillig *et al.*, 2019) and against (Ryan & Graham, 2002), particularly when criteria for AMF benefits are defined either as

yield or sustainability. In this regard, our modelling study confirms the strong and long term influence of AMF to plant biomass and productivity for environments where nutrients are the most limiting factor, but weaker effects where they are not. Second, the shift in both SRL and SLA distributions when AMF is deactivated suggest that plant communities may differ significantly when mycorrhizal symbiosis is absent, particularly towards conservative, conservative assemblages. Our model is thus in line with suggestions from ecosystem restoration practices: the introduction of AMF inoculation can both

impact plant growth and community composition (Lin *et al.*, 2015). In order to better explore from a model perspective AMF influence to the whole ecosystem, the implementation and evaluation through simulated experiments of several key processes such as soil aggregation, seedling survival, resistance to pathogens and resistance to invasive species as well as a thorough analysis of AMF effects to soil stocks and fluxes, would be invaluable (Rillig *et al.*, 2019).

## 6.  Conclusions

Even though fine-root traits and AMF are strongly linked and thought to impact ecosystem functioning, variation in fine-root traits has hardly been addressed in process-based DVMs and C cycle models. Here, we present a parsimonious, generalized approach for implementing fine-root trait diversity and the fungal collaboration gradient in a widely used DVM. With local data from a tropical montane biodiversity hotspot, we show that the model can reproduce fine-root traits, AMF colonization and aboveground vegetation features along a nutrient-limitation gradient. Model results confirm an expected crucial role of

AMF for nutrient uptake and vegetation productivity at the high elevation, strongly nutrient-limited site. The model also reveals potential linkages between fine-root traits, AMF colonization and aboveground vegetation traits. We also show how trait-based DVMs can be a powerful tool for testing ecological hypotheses concerning complex interactions in ecosystems. Future research and model development, however, should focus more on belowground traits and their interaction with mycorrhiza and aboveground traits. Belowground processes and nutrient dynamics might, in the future, become even more important because plants will be even more important due to enhanced photosynthetic activity as atmospheric $CO_2$ concentrations continue to rise.

## 7. Appendix A

### Evaluating carbon transfer from plant to fungi

Increasing transfer of C from plants to fungi reduced nutrient limitation in our simulations (Figs. A3e-f), total vegetation biomass and productivity (Figs. 3a-b). The analysis of how the average fine-root traits (SRL and AMF colonization) changed with increasing maximum C costs, $r_{max}$ (Figs. 3c and A3a) reveals a clear cost-benefit pattern in which too little or too much C transfer, here also related to fungal colonization, is detrimental to plant productivity. Too little C transfer ($r_{max} < 0.5$) resulted in low total AMF biomass (Fig. A3b). For $r_{max}$ values above 0.5, individuals with this high C transfer and higher AMF colonization rates are outcompeted, driving down community average AMF colonization rates (Fig. 3c). This occurs since the increasingly large AMF mass these plants would need to support offer no further benefit in nutrient limitation alleviation. The resulting AMF simulated biomass from an $r_{max}$ of 0.5 and above (between around 0.35 and 0.45 MgC ha$^{-1}$, Fig. A3b) fell within the 0.054 - 0.9 Mg C ha$^{-1}$ of observations (Parihar *et al.*, 2020). Since AMF biomass is an emergent property of the model (i.e. not prescribed), this fit within observations indicates that our assumptions and prescribed parameters (such as AMF turnover) produce a realistic representation of AMF function.

| Variable | Description | Unit | Value(s) | Reference |
|---|---|---|---|---|
| *Traits* | | | | |
| sla_min* | minimum randomized specific leaf area | cm² g$^{-1}$ | 15.5 | (Báez & Homeier, 2018) |
| sla_max* | maximum randomized specific leaf area | cm² g$^{-1}$ | 273.5 | (Báez & Homeier, 2018) |

| | | | | |
|---|---|---|---|---|
| wsg_min* | minimum randomized wood specific gravity | g cm$^{-3}$ | 0.158 | (Báez & Homeier, 2018) |
| wsg_max* | maximum randomized wood specific gravity | g cm$^{-3}$ | 1.02 | (Báez & Homeier, 2018) |
| srl_min* | minimum randomized specific root legth | cm g$^{-1}$ | 191.2 | (Pierick et al. 2021) |
| srl_max* | maximum randomized specific root legth | cm g$^{-1}$ | 11,712.80 | (Pierick et al. 2021) |
| rmax | fraction of individual´s NPP allocated to growth to be transferred to AMF at 100% colonization rate | - | 0.5 | this study |
| longevity_myco | AMF longevity | d | 80 | (Raven et al. 2018) |
| cton_myco | AMF C:N ratio, used for calculating respiration. | - | 15 | (Orwin at al. 2011) |
| shl_myco | Fixed specific hyphal length for AMF | cm g$^{-1}$ | 1.33 10$^{8}$ | (Raven et al. 2018) |
| d_myco | hyphae diameter | cm | 2 10$^{-4}$ | (Raven et al. 2018) |

*Michaelis-Menten kinetics*

| | | | | |
|---|---|---|---|---|
| nuptoroot | Max. inorganic N (Ni = $NH_4$+NO3-) mass uptake per fine-root C mass per day | gN gC$^{-1}$ d$^{-1}$ | 3.57 10$^{-3}$ | (Rothstein *et al.*, 2000) |
| puptoroot | Max. $PO_4$ mass uptake per fine-root C mass | gP gC$^{-1}$ d$^{-1}$ | 5.9 10$^{-4}$ | (Silveira & Cardoso, 2004) |

per day

| | | | | |
|---|---|---|---|---|
| KmN_volume | Half saturation concentration for Ni uptake | kg l$^{-1}$ | 2.1 10$^{-6}$ | (Rothstein *et al.*, 2000) |
| KmP_volume | Half saturation concentration for PO$_4$ uptake | kg l$^{-1}$ | 2.15 10$^{-7}$ | (Silveira & Cardoso, 2004) |
| nuptomyco | Max. inorganic N (Ni = NH$_4$+NO3-) mass uptake per amf C mass per day | gN gC$^{-1}$ d$^{-1}$ | 7 10$^{-3}$ | (Pérez-Tienda *et al.*, 2012) |
| puptomyco | Max. PO$_4$ mass uptake per fine AMF C mass per day | gP gC$^{-1}$ d$^{-1}$ | 9.13 10$^{-4}$ | (Silveira & Cardoso, 2004) |
| KmN_volume _myco | Half saturation concentration for AMF Ni uptake | kg l$^{-1}$ | 4.5 10$^{-9}$ | (Pérez-Tienda et al., 2012) |
| KmP_volume_ myco | Half saturation concentration for AMF PO$_4$ uptake | kg l$^{-1}$ | 1.63 10$^{-7}$ | (Silveira & Cardoso, 2004) |

**Table A1. Model parameters relevant to the new implementations of LPJ-GUESS-NTD. Variables with asterisks were measured in our study sites.**


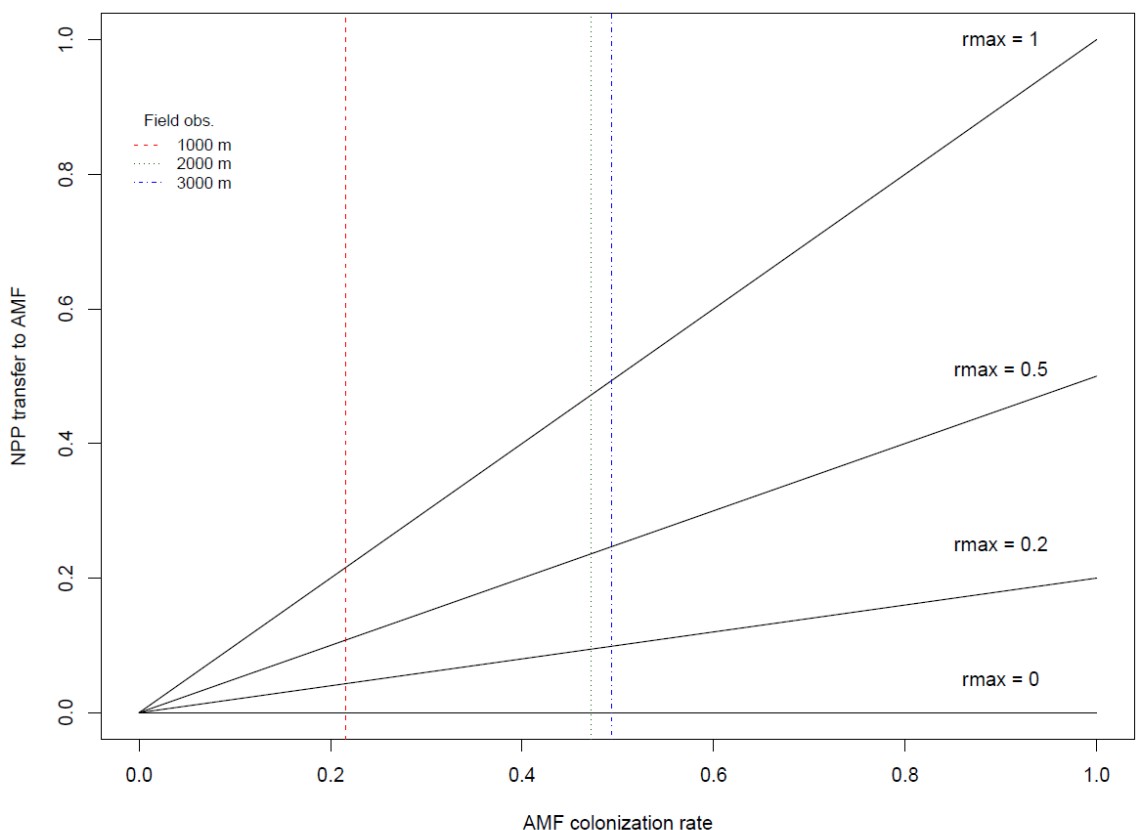

**Figure A1. Relationship between the $r_{max}$ parameter and NPP transfer to mycorrhiza.**


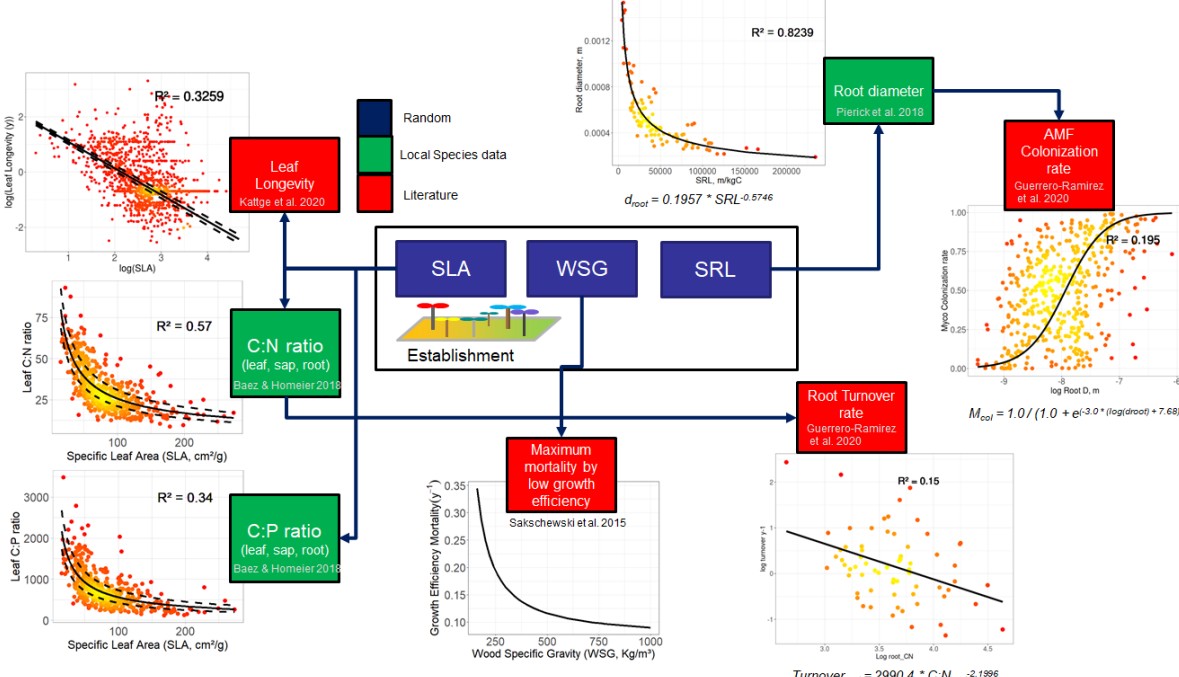

**Figure A2. Complete scheme of model trait-trait relationships (trade-offs) after the inclusion of the fine-root traits and collaboration gradient. Traits in blue boxes are randomized uniformly at establishment according to a fixed range. Green boxes represent traits whose trade-offs were defined using measurements only from our study site, and red boxes were defined using data from the literature. SLA: specific leaf area, WSG: wood specific gravity; SRL: specific root length.**


|  | 1000 | | 2000 | | 3000 | |
|---|---|---|---|---|---|---|
|  | **AMF-on** | **AMF-off** | **AMF-on** | **AMF-off** | **AMF-on** | **AMF-off** |
| **SRL** | 2691.32 | 3068.88 | 2299.13 | 3820.19 | 1035.64 | 4482.34 |
| **SLA** | 36.37 | 38.71 | 38.90 | 29.60 | 37.96 | 21.86 |

Table A2. Median Absolute Error measures between the scenarios (AMF-on and AMF-off) and field data for the three elevation sites.

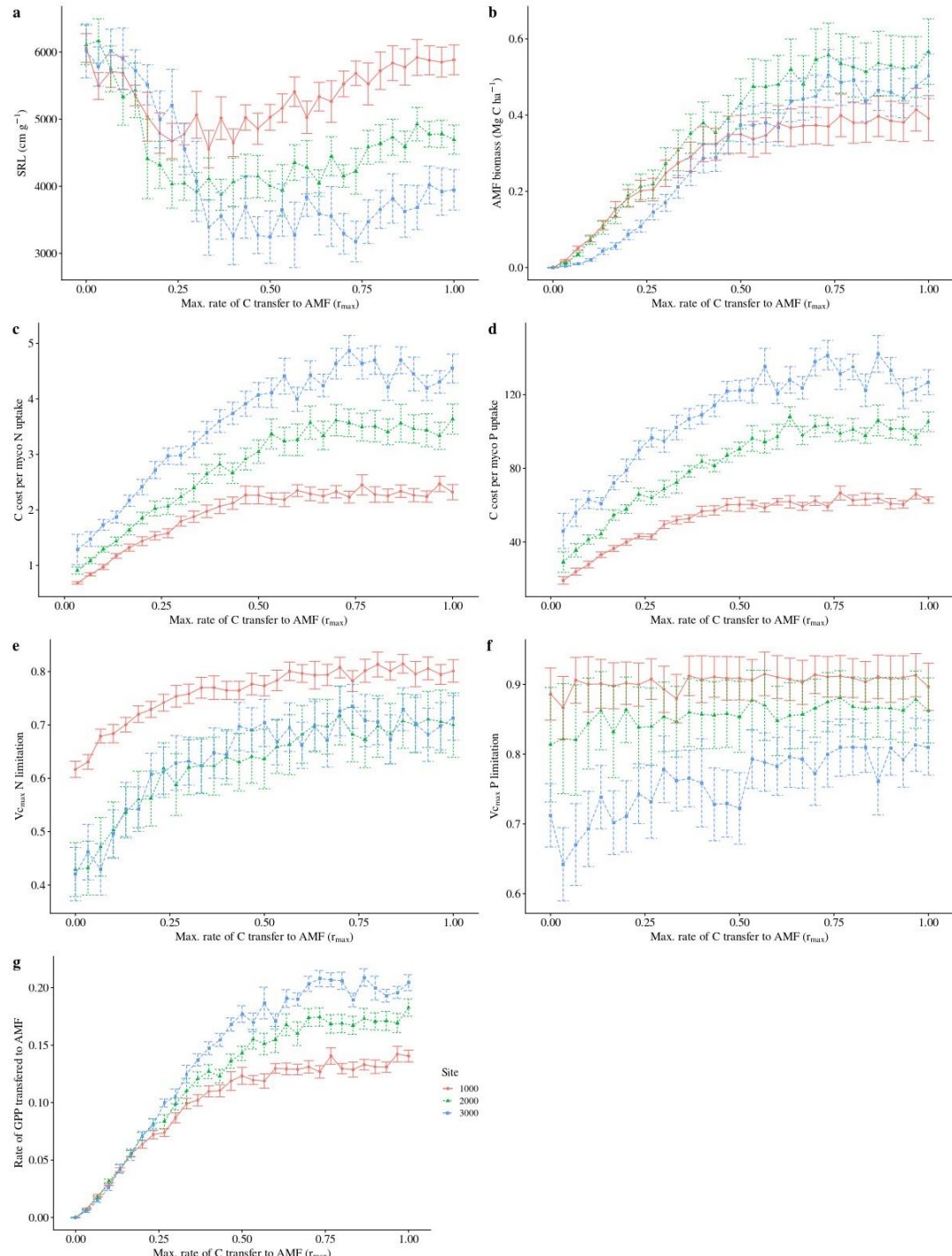

**Figure A3. Sensitivity analysis of the parameter $r_{max}$, maximum C allocation to growth for mycorrhiza. Each point represents 200 year averages using a particular $r_{max}$. Whiskers indicate +/- SD from the 200 years of each run. Red lines: 1,000 m. Green lines: 2,000 m. Blue lines: 3,000 m.**

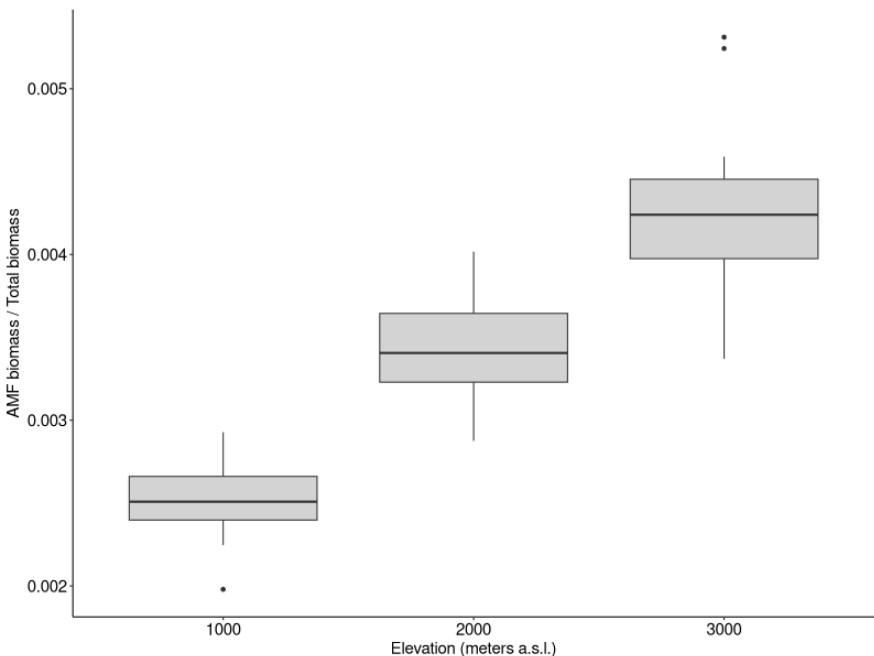

**Figure A4. Simulated proportion of AMF to total vegetation biomass for the three elevation sites.**


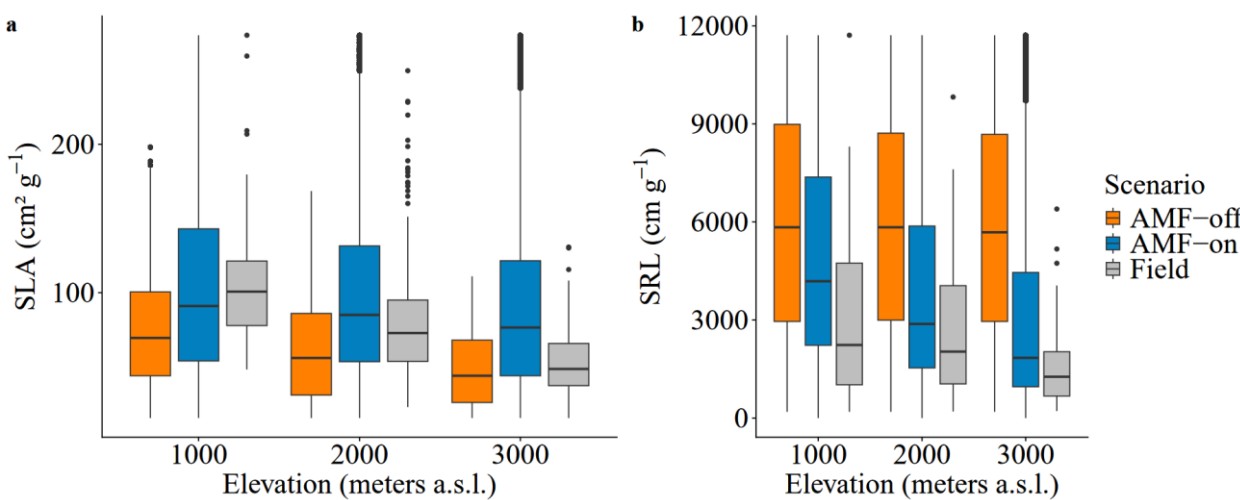

Figure A5. Boxplots of a. specific leaf area (SLA) and b. specific root length (SRL) for the two simulation scenarios (AMF-on and AMF-off) and field measurements.

## 8. Data availability

All field data referenced and collected for this work is available for download at the FOR2730 data warehouse which is accessible at http://vhrz669.hrz.uni-marburg.de/tmf_respect/data_pre.do?cmd=showall. Simulation results, and plotting script for all figures are accessible at https://zenodo.org/records/13772012.

## 9. Author contribution

Mateus Dantas de Paula: design of the research; performance of the research; data analysis and interpretation; writing the manuscript

Tatiana Reichert: performance of the research; data analysis and interpretation; writing the manuscript

Laynara F. Lugli: performance of the research; data analysis and interpretation; writing the manuscript

Erica McGale: performance of the research; writing the manuscript

Kerstin Pierick: performance of the research; data analysis and interpretation; writing the manuscript

João Paulo Darela Filho: performance of the research; data analysis and interpretation; writing the manuscript

Liam Langan: data analysis and interpretation; writing the manuscript

Jürgen Homeier: performance of the research; data analysis and interpretation; writing the manuscript

Anja Rammig: performance of the research; data analysis and interpretation; writing the manuscript

Thomas Hickler: design of the research; performance of the research; data analysis and interpretation; writing the manuscript

## 10. Competing interests

The authors declare that they have no conflict of interest.

## 11. Acknowledgements

We are grateful to the DFG (German Research Foundation) for funding this study in the scope of the Research Unit FOR2730 RESPECT. We thank the Ecuadorian Ministry of Water and the Environment (MAAE) for the permission to conduct research, and to the foundation Nature and Culture International (NCI) for logistical support. LFL acknowledges the Bavarian State Chancellery (Project Amazon-FLUX).

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
