# Peer review of "The fungal collaboration gradient drives root trait distribution and ecosystem processes in tropical montane forests"

_EGUsphere, 2024_

## Referee Comment (RC2)

Dear Reviewer,
Thank you very much for this insightful and extensive review. Below are the descriptions of all addressed points.

**The authors have addressed many of this reviewer's concerns. However, by providing more detailed information on vegetation types, investigating the ecological mechanisms underlying biomass differences, and further examining the broader ecological implications of their findings, the authors can substantially enhance the quality and impact of their work. Specific comments are provided below.**

Specific comments:
Abstract (Section 1):

The acronym AMF should be spelled out on its first mention (line 22).
The acronym has been now described in L20, many thanks.

While the abstract mentions "linkages between below- and aboveground traits" (line 25), elaborating on the nature or significance of these associations would better inform readers of their ecological implications.
On L26 it was added that "AMF promotes more acquisitive leaf traits".

Numerical results on SRL and AMF colonization along the altitudinal gradient are missing. Including key statistical results would add weight to the claims.
Thank you, a new table has been added (Table 1) with all numerical results, including SRL and AMF colonization. This has also been added to the abstract.

Although specific to a tropical montane forest in southern Ecuador, expanding on the study's contributions to global ecological or climate models would strengthen its impact.
Thank you for the suggestion, in our last line of the abstract we suggest that the study's approach can be applied to other regions as well.

Introduction (Section 2):

The introduction provides a solid background on belowground processes, fine root characteristics and the role of mycorrhizae in ecosystem functioning. The references are diverse and robust, effectively highlighting the novelty in

modelling of modelling fine root characteristics and FCG. However, the section could benefit from improved clarity and flow:

Simplifying dense sentences:

- Lines 32-33 could be rephrased as, e.g.: "Soils at depths of up to 200 cm store an estimated 2400 Pg C globally, highlighting their importance in the carbon cycle (Batjes, 1996). This is nearly nine times the amount stored in global forests (Santoro et al., 2021)".
Thank you for this suggestion, sounds much better, this has been included in the text.

- Lines 38-39 could be simplified to, e.g.: "Fine root traits - such as branching patterns, root depth and diameter - play a critical role in nutrient and water uptake. These traits may also shape species coexistence in specific environments (Nie et al., 2013)".
This was nicely noted – the text was slightly changed to add community composition as: "Fine root traits - such as branching patterns, root depth and diameter - play a critical role in nutrient and water uptake. These traits may also shape species coexistence, and thus community composition in specific environments"

Restructuring dense paragraphs:

- The paragraph in lines 47-53 could be condensed to, e.g.: "Advances in belowground phenotyping have enabled researchers to synthesize fine-root traits within the global spectrum of plant form and function (Weemstra et al., 2016, 2022; Weigelt et al., 2021). Similar efforts have been made for leaf and wood traits (Wright et al., 2004; Chave et al., 2009). This framework highlights how trade-offs in physiological and morphological traits influence species coexistence (Shipley et al., 2006). By analyzing the co-occurrence of plant traits, researchers have identified new trade-off gradients (Guerrero-Ramírez et al., 2020; Kattge et al., 2020)".
Thank you for this suggestion, which has been accepted.

- Trade-offs and gradients appear to be over-explained in lines 53-59, and could be condensed and focused to, e.g.: "Plant traits often exhibit trade-offs, such as the conservation gradient seen in leaves, where traits range from high productivity to high longevity (Wright et al., 2013; Díaz et al., 2016). Similarly, root traits show trade-offs, though their patterns differ from those observed in leaves (Carmona et al., 2021)".

The suggestion was accepted with the following changes: "By analyzing the co-occurrence of plant traits, researchers have identified relationships between them and developed the concept of the global spectrum of plant form and function (Díaz *et al.*, 2016; Guerrero-Ramírez *et al.*, 2020; Kattge *et al.*, 2020)This includes the leaf and wood economics spectrum, where stoichiometric traits (e.g. leaf C:N) are related to high productivity or high longevity strategies, in a "conservation" or "productivity" trade-off axis (Chave *et al.*, 2009; Wright *et al.*, 2013; Díaz *et al.*, 2016). This concept highlights how trade-offs in physiological and morphological traits influence species coexistence (Shipley *et al.*, 2006).. Advances in belowground phenotyping have enabled researchers to synthesize fine-root traits within the global spectrum of plant form and function (Weemstra *et al.*, 2016, 2022; Weigelt *et al.*, 2021). Fine root stoichiometry traits were also observed to produce such a conservation gradient, however root morphological traits such as root diameter did not seem to align with the existing conservation axis (Carmona *et al.*, 2021). ", many thanks.

- Lines 67-69 appear overly complex and could be simplified and broken down, e.g.: "Although plants must transfer carbon to fungi as part of their partnership, the fungi's extensive hyphae networks significantly boost nutrient and water absorption. This collaboration offers thick-rooted plants an alternative strategy to relying solely on fine roots (Kakouridis et al., 2022)".
Thank you for this suggestion, it has been accepted.

   Improving transition:

- The sentence in line 60 should belong to the previous paragraph, followed by a sentence of the type "Capturing such dynamics is crucial for process-based dynamic vegetation models (DVMs), which rely on generalized ecological representations to simulate plant and soil processes". This means that the concepts currently expressed from line 75 onwards should appear earlier to ensure a smooth transition between topics like fine root traits, fungal collaboration gradients and DVMs.
Thank you, but we believe the definition of the fungal collaboration gradient should in fact come after the general root trait spectrum description, and the introduction on DVMs should come close to the end.

- Lines 78-82 could also be smoothed, e.g. "Aboveground plant traits are often analyzed using the 'leaf economics spectrum,' a framework that classifies leaves based on a trade-off between rapid growth and resource conservation (Wright et al., 2004). Including such frameworks in DVMs has provided insights into nutrient dynamics and community resilience (Sakschewski et al., 2015, 2016; Dantas de Paula et al., 2021). By contrast, belowground processes, despite their

critical ecological roles, remain underrepresented in these models (Langan et al., 2017; Sakschewski et al., 2021)".

Thanks for the suggestion, this part has been amended as follows: "Aboveground plant trait variation has been implemented using the 'leaf economics spectrum,' a framework that classifies leaves based on a trade-off between rapid growth and resource conservation (Wright *et al.*, 2004). Including such frameworks in DVMs has provided insights into nutrient dynamics and community resilience (i.e., Sakschewski *et al.*, 2015, 2016; Dantas de Paula *et al.*, 2021). By contrast, belowground processes, despite their critical ecological roles, remain underrepresented in these models (Langan *et al.*, 2017; Sakschewski *et al.*, 2021)"

- Similarly, to ensure a smooth transition from DVM limitations to study objectives, the authors might consider something like "Existing models simplify or omit critical variations in root traits and mycorrhizal dynamics, limiting their ability to capture site-specific belowground processes. To address these gaps, this study develops a dynamic approach that integrates detailed root trait and fungal interaction data into DVMs for broader ecological applications".

Thank you, the last line was added to the last paragraph of the introduction, which is where our specific approach is described.

   Explaining concepts:

- The authors could further describe the "fungal collaboration gradient" (line 60) as a spectrum of strategies that plants use to interact with mycorrhizal fungi: at one end, plants invest heavily in root traits that enhance independent nutrient uptake, while at the other end they rely more on fungal partners to exchange nutrients for carbon.

Thank you, a line related to this description was added at the end of paragraph of line 60. "A spectrum of strategies thus emerges: at one end, plants invest heavily in root traits that enhance independent nutrient uptake, while at the other end they rely more on fungal partners to exchange nutrients for carbon."

- Similarly, the authors can briefly introduce and support DVMs as simulation tools that predict how plant communities respond to environmental changes by incorporating simplified representations of ecological processes, such as growth, nutrient cycling and competition.

Thanks for the suggestion, but we believe the DVM concept is already explained enough here at line 75, and more details in the methods.

   Introducing hypotheses:

The current introduction does include hypotheses (from line 98 onwards), but they can be made more explicit, with a clear framing sentence following the rationale for focusing on root traits and fungal interactions. For example: "In this study, we hypothesize that incorporating the FCG into a trait-based DVM will improve predictions of root trait distributions, biomass, and productivity across nutrient gradients. Specifically, we predict that ...". This approach would make the goals and hypotheses prominent and easier to follow.

Thank you for the comment. In the last few lines of the last paragraph, the hypothesis as suggested were made more clear:

"We aim here to test with our model implementation the general hypothesis that the FCG is an important factor behind the observed root trait distribution, forest biomass and productivity. , More specifically, we hypothesize that (1) in line with the mycorrhizal colonization gradient and field measurements of root traits, as available nutrients decrease with elevation, simulated community average values of SRL decrease, root diameters increase and colonization rates by AMF increase when the FCG is active. Next, we removed the mycorrhizal fungi in a simulated exclusion experiment, and (2)expect that in the absence of AMF, plant biomass and productivity would be affected, and SRL between the different sites of the elevation gradient would not differ. In other words, this latter result would imply that AMF drives morphological root diversity."

   Synthesizing related studies:

While the introduction includes citations from relevant and recent studies on several topics, i.e. fine root traits (e.g. Nie et al., 2013; Bardgett et al., 2014), mycorrhizal interactions (e.g. Van Der Heijden et al., 2008; Hawkins et al., 2023), modeling approaches and limitations (e.g., Langan et al., 2017; Dantas de Paula et al., 2021), a broader review of (a) root-mycorrhizal interactions in different ecosystems (e.g., temperate or boreal forests) for comparative insights, and (b) previous attempts to incorporate belowground traits into models (even outside of DVMs), would contextualize the novel contributions of this study.

 More detail on different plant-mycorrhizal interactions and modelling approaches are very interesting prospects to include, and this has been done to a certain extent in the discussion – however we feel that these points should be included in further publications, where our model is extended to other environments.

Material and methods (Section 3):

The Materials and Methods section is generally well-written, providing a solid outline of the model's structure and integration of the FCG. However, certain aspects require clarification or expansion to ensure transparency and reproducibility. Specifically, a dedicated sub-section (e.g. "3.1. Land use and vegetation cover") should be created to include detailed information about historical and current land use, vegetation types, conservation status, and anthropogenic impacts. The current description does not specify the vegetation type at each elevation site (e.g., primary forest, secondary forest, or disturbed areas), which is crucial because vegetation types significantly influence nutrient cycling, organic matter decomposition, and fungal associations. It remains unclear whether the sites are pristine or subject to human activity. For instance: Are these sites located within protected areas? Do they feature continuous forest cover, fragmented landscapes, or mixed land use? The authors should describe the dominant vegetation types or species at each elevation, such as tropical montane cloud forests, mixed forests, or other specific plant communities. A brief mention of historical land use (e.g., deforestation or reforestation efforts) would provide context for the current ecosystem conditions.

Thank you for pointing this out. The study sites are located within a national park, and are all considered to be primary forest. This has been added to the text: "These three sites are considered to be primary forest remnants 🗎 are located within the Podocarpus National Park, where In this location, a wealth of biotic and abiotic measurements have been carried out since the early 2000's (Beck et al., 2008; Bendix et al., 2013, 2021)." Further description of vegetation types, plant community composition, land use in the surroundings, etc. are all thoroughly described in these two references and may lead to an excessive word count in the manuscript.

This considered, the following changes are recommended to improve organization and clarity:

Section 3.1 (to be renumbered 3.2 under a new title, e.g. "Abiotic characteristics and nutrient gradients") could benefit from including quantitative data on environmental factors, such as soil nutrient levels and moisture content, and elaborate on how these variables were incorporated into the model.
Thank you for the suggestion, but we have already included in this section measurements of mineral nutrient content, and moisture is not a limiting factor for these three sites.

Section 3.2 (to be renumbered 3.3) could anticipate a discussion on specific algorithms and assumptions used to simulate root growth dynamics and

mycorrhizal interactions, which would add transparency to the mechanics of the model.

Thank you for the comment, the following text has been added: "This previous implementation therefore did not provide a realistic relationship between C invested into nutrient acquisition (as roots or mycorrhizae) and effective acquisition capacity. Although root biomass in general terms translates into higher nutrient or water uptake capacity, there is high variation for same biomass values, and this is thought to result from root morphological trait variation (Kokko et al., 1993). Even though root architecture (i.e. root distribution along different soil layers) and rooting depth are considered in our and other models (Langan et al., 2017; Sakschewski et al., 2021), to our knowledge morphological traits such as root diameter and specific length have not been implemented."

Section 3.3 (to be renumbered 3.4) could expand on the interactions between the FCG and external drivers (e.g., climate, competition, disturbances) to provide a more comprehensive view of their role in ecosystem dynamics.

Thanks for this, the following lines were added to this section: "We expect then that in more nutrient limited environments, where C is not limiting and light is abundant, such as in higher elevation areas, mycorrhiza colonization will be high. On the other hand, in more C limited environments such as lower elevation sites (i.e. where light competition plays a large role) investment in mycorrhiza will not lead to higher fitness, leading to lower colonization rates."

Section 3.4 (to be renumbered 3.5) could include a more detailed explanation of how the datasets used were used. This could include descriptions of the variables measured and how they were incorporated into the model to improve transparency and reproducibility.

For this we described better the N and P deposition data, and added references for more details on how the driver data influences the results. The added text is: Soil data and parameters were taken from the World Soil database (FAO/IIASA/ISRIC/ISS-CAS/JRC, 2012); current N and P deposition rates used were from Dantas de Paula et al., (2021); which were measured weekly during the same 1999 – 2018 period. Details on how driver data influences ecosystem processes can be found at (Smith et al., (2014). and (Dantas de Paula et al., (2021).

Section 3.5 (to be renumbered 3.6) could enhance the ecological relevance of the AMF-on and AMF-off scenarios by linking them to practical applications (e.g., conservation strategies and ecosystem management).

Thank you for the comment, we believe  practical applications are better suited for the discussion.

Section 3.6 (to be renumbered 3.7) could provide a more detailed statistical analysis, including performance indicators like root mean square error (RMSE) or Nash-Sutcliffe Efficiency (EF), to assess the model accuracy.
Thank you for this. Since both the field and simulation data have a lot of outliers, the median absolute error was used to estimate the model performance against field data. The values were added to a new table in the appendix, Table A2. In addition, a new figure was added (Figure A5) in which SRL and SLA are added as boxplots also considering the field data, in order to improve interpretation of the results. The following lines were added to section 4.2: "Model performance thus improved in the AMF-on scenario, reducing median absolute errors for SRL in comparison with the AMF-off. SLA on the other hand was slightly worse with the activation of AMF uptake (Table A2, Figure A5)."
Section 3.7 (to be renumbered 3.8) could discuss the ecological implications of the sensitivity analysis findings, particularly highlighting their relevance to management practices and identifying areas for future research.
Thank you for the suggestion. Please check the new additions in the discussion section for more details on this.

Results (Section 4):

In order to improve the transfer of this study to practical applications, a new table was added (Table 1) which includes simulated data on soil C stocks, N mineralization rates and litter stoichiometry. These new data show how the presence of AMF significantly influence soil structure and dynamics. Discussion points on this were also included.

Section 4.1 could elaborate on the ecological significance and practical applications of the optimal rmax value (0.5). For example, how does varying rmax inform decisions on ecological restoration or nutrient cycling management?
Thank you, this has been added in the discussion (L 502)

Section 4.2 could provide details on the statistical significance of observed patterns (e.g., p-values) and clarify whether the AMF-off scenario exhibited an exact uniform distribution or simply a lack of observed patterns. In addition, the paragraph on SLA traits under the AMF-off scenario requires further explanation of the ecological implications. What does a "conservative" (lines 308) trait like lower SLA mean for plant strategy or ecosystem processes? Further discussion of these traits in terms of resource allocation or plant adaptation would strengthen the argument.

Thank you. We also added that the AMF-off scenario has a lack of trends regarding SRL, and ran a t-test for the AMF scenarios along all variables, see Table 1 legend. For more details regarding this comment, please check the discussion.

Section 4.3 requires further interpretation of the reduction in biomass at the 3,000 m site (80.6%) under the AMF-on scenarios. A brief discussion of the ecological mechanisms behind this pattern would enrich the results and help to explain why such a large reduction in biomass occurs at higher altitudes (possibly in terms of nutrient availability, plant strategy or other environmental factors).
Please check the discussion for more details on this.

Discussion (Section 5):

The discussion briefly touches on potential ecological mechanisms, in particular nutrient limitation and mycorrhizal requirements at higher elevations. However, there is limited discussion of the biological mechanisms driving the large biomass differences observed in the AMF-on scenario, particularly at higher elevations. As noted above, the section should expand on the biological mechanisms underlying biomass differences, particularly at higher elevations, and explore interactions between FCG and other factors (e.g., climate, competition, disturbance).

Conclusion (Section 6):

The conclusion could be strengthened by addressing interactions between the FCG and other factors, such as climate or soil type. While the authors acknowledge the need for future research on belowground traits and their interactions with AMF and aboveground traits, the discussion could be expanded to include how this understanding can be applied to ecosystem management and conservation practices.
Thank you for suggesting this. As noted in several previous comments, links to conservation practices, management and restoration would greatly enrich the discussion. Here we have added a whole paragraph relating to this in section 5.4, which reads:
Finally, our study can provide important insights into the recent discussion on the importance of AMF for ecosystem management, in spite of its limitations.

First, the relevance of AMF abundance and diversity for agriculture has been argued for (Rillig et al., 2019) and against (Ryan & Graham, 2002), particularly when criteria for AMF benefits are defined either as yield or sustainability. In this regard, our modelling study confirms the strong and long term influence of AMF to plant biomass and productivity for environments where nutrients are the most limiting factor, but weaker effects where they are not. Second, the shift in both SRL and SLA distributions when AMF is deactivated suggest that plant communities may differ significantly when mycorrhiza is absent, particularly towards conservative, ruderal assemblages as can be found in many degraded environments. Our model is thus in line with suggestions from ecosystem restoration practices: the introduction of AMF inoculation can both impact plant growth and community composition (Lin et al., 2015). In order to better explore from a model perspective AMF influence to the whole ecosystem, the implementation and evaluation through simulated experiments of several key processes such as soil aggregation, seedling survival, resistance to pathogens and resistance to invasive species as well as a thorough analysis of AMF effects to soil stocks and fluxes, would be invaluable (Rillig et al., 2019).

The implications of rising atmospheric CO2 on belowground processes and mycorrhizal interactions (lines 478-479) also warrant further development. Thank you, a lines and reference on this were added at L 480, where AMF turnover is crucial for estimation of CO2 effects on mycorrhiza, as well as in L 530.

In addition, exploring the potential for extending this approach to other ecosystems, such as temperate forests, grasslands or drylands, would increase the impact of the study.
==Thank you, we have added extra lines on including Ectomycorrhiza fungi and exploring this approach for estimating global patterns of mycorrhiza types.==
[Figure]

Finally, a more explicit link between the results of the study and broader ecological theory would strengthen the conclusions. This could include discussion of how the integration of root traits and mycorrhizal cooperation into DVMs advances our understanding of plant-soil feedback mechanisms, nutrient cycling and ecosystem resilience.
==Thanks for the suggestion, this is now explored in more detail in combination with the discussion on the soil C stocks, net nutrient mineralization and litter stoichiometry results .==

Appendix A (Section 7):

The appendix significantly enhances the transparency and reproducibility of the study by providing additional information, including data tables and analyses that help interpret the model results. However, the authors may consider omitting the heading 7.1, as it pertains to a single section.

Thank you for the suggestion, the heading has been now omitted.

---

## Author Response (AR1)

*Note: The line numbering of the reviewer's comments refer to the unrevised manuscript version (Dantas_de_Paula_et_al_2024_Biogeosciences_Main_Document) and the line numbering of the replies refer to the revised manuscript version with tracked changes (Dantas_de_Paula_et_al_2024_Biogeosciences_Main_Document_reviewed_tracked_v1).*

Authors replies are in bold and italic.

**REVIEWER 1, 1st comment**

General comment:

The manuscript presents a comprehensive study on the interactions between arbuscular mycorrhizal fungi (AMF) and root traits in a tropical montane forest ecosystem. Using a dynamic vegetation model (DVM), the authors explore how these interactions influence aboveground vegetation traits and ecosystem processes. This research is timely and highly relevant, given the increasing interest in understanding both above- and below-ground interactions in the context of global change.

The study significantly advances our understanding of the role of AMF in shaping plant communities by integrating a novel "fungal collaboration gradient" (FCG) within a trait-based DVM. This approach enables nuanced exploration of how root traits and mycorrhizal associations, particularly AMF, influence ecosystem dynamics in tropical montane forests.

The inclusion of empirical data on specific root length (SRL) and AMF colonization across an altitudinal gradient enhances the model's credibility and relevance. One key finding is the significant impact of deactivating AMF-mediated nutrient uptake, with biomass values potentially reduced by up to 80%, emphasizing the critical role of mycorrhizal associations in nutrient acquisition.

While the results are compelling, the discussion would benefit from deeper integration with existing literature, particularly through comparisons with previous studies on AMF and root traits. Additionally, exploring the practical applications of the findings, particularly for ecosystem management, would strengthen the study's relevance.

In summary, the manuscript is well-conceived and addresses a significant gap in ecological research. However, revisions are needed to improve clarity, structure, and depth of analysis. Given the scope of the requested revisions—reorganization, additional data, expanded literature integration, methodological elaboration, and enhanced analysis—this report constitutes a major revision request. Addressing these revisions aims to deepen the manuscript's analysis and elevate its scientific standards prior to publication.

*Thank you very much for this insightful and extensive review. Below are the descriptions of all addressed points.*

Specific comments:

Abstract (Section 1):

The acronym AMF should be spelled out on its first mention (line 22).

*The acronym has been now described in L20, many thanks.*

While the abstract mentions "linkages between below- and aboveground traits" (line 25), elaborating on the nature or significance of these associations would better inform readers of their ecological implications.

*On L26 it was added that "AMF promotes more acquisitive leaf traits" – more details are not possible unfortunaltely here because of abstract word constraints.*

Numerical results on SRL and AMF colonization along the altitudinal gradient are missing. Including key statistical results would add weight to the claims.

*Thank you, a new table has been added (Table 1) with all numerical results, including SRL and AMF colonization. This has also been added to the abstract.*

Although specific to a tropical montane forest in southern Ecuador, expanding on the study's contributions to global ecological or climate models would strengthen its impact.

*Thank you for the suggestion, in our last line of the abstract we suggest that the study's approach can be applied to other regions as well.*

Introduction (Section 2):

The introduction provides a solid background on belowground processes, fine root characteristics and the role of mycorrhizae in ecosystem functioning. The references are diverse and robust, effectively highlighting the novelty in modelling of modelling fine root characteristics and FCG. However, the section could benefit from improved clarity and flow:

Simplifying dense sentences:

- Lines 32-33 could be rephrased as, e.g.: "Soils at depths of up to 200 cm store an estimated 2400 Pg C globally, highlighting their importance in the carbon cycle (Batjes, 1996). This is nearly nine times the amount stored in global forests (Santoro et al., 2021)".

***Thank you for this suggestion, sounds much better, this has been included in the text.***

- Lines 38-39 could be simplified to, e.g.: "Fine root traits - such as branching patterns, root depth and diameter - play a critical role in nutrient and water uptake. These traits may also shape species coexistence in specific environments (Nie et al., 2013)".

***This was nicely noted – the text was slightly changed to add community composition as: "Fine root traits - such as branching patterns, root depth and diameter - play a critical role in nutrient and water uptake. These traits may also shape species coexistence, and thus community composition in specific environments"***

Restructuring dense paragraphs:

- The paragraph in lines 47-53 could be condensed to, e.g.: "Advances in belowground phenotyping have enabled researchers to synthesize fine-root traits within the global spectrum of plant form and function (Weemstra et al., 2016, 2022; Weigelt et al., 2021). Similar efforts have been made for leaf and wood traits (Wright et al., 2004; Chave et al., 2009). This framework highlights how trade-offs in physiological and morphological traits influence species coexistence (Shipley et al., 2006). By analyzing the co-occurrence of plant traits, researchers have identified new trade-off gradients (Guerrero-Ramírez et al., 2020; Kattge et al., 2020)".

***Thank you for this suggestion, which has been accepted.***

- Trade-offs and gradients appear to be over-explained in lines 53-59, and could be condensed and focused to, e.g.: "Plant traits often exhibit trade-offs, such as the conservation gradient seen in leaves, where traits range from high productivity to high longevity (Wright et al., 2013; Díaz et al., 2016). Similarly, root traits show trade-offs, though their patterns differ from those observed in leaves (Carmona et al., 2021)".

*The suggestion was accepted with the following changes: "By analyzing the co-occurrence of plant traits, researchers have identified relationships between them and developed the concept of the global spectrum of plant form and function (Díaz et al., 2016; Guerrero-Ramírez et al., 2020; Kattge et al., 2020)This includes the leaf and wood economics spectrum, where stoichiometric traits (e.g. leaf C:N) are related to high productivity or high longevity strategies, in a "conservation" or "productivity" trade-off axis (Chave et al., 2009; Wright et al., 2013; Díaz et al., 2016). This concept highlights how trade-offs in physiological and morphological traits influence species coexistence (Shipley et al., 2006).. Advances in belowground phenotyping have enabled researchers to synthesize fine-root traits within the global spectrum of plant form and function (Weemstra et al., 2016, 2022; Weigelt et al., 2021). Fine root stoichiometry traits were also observed to produce such a conservation gradient, however root morphological traits such as root diameter did not seem to align with the existing conservation axis (Carmona et al., 2021). ", many thanks.*

- Lines 67-69 appear overly complex and could be simplified and broken down, e.g.: "Although plants must transfer carbon to fungi as part of their partnership, the fungi's extensive hyphae networks significantly boost nutrient and water absorption. This collaboration offers thick-rooted plants an alternative strategy to relying solely on fine roots (Kakouridis et al., 2022)".

*Thank you for this suggestion, it has been accepted.*

Improving transition:

- The sentence in line 60 should belong to the previous paragraph, followed by a sentence of the type "Capturing such dynamics is crucial for process-based dynamic vegetation models (DVMs), which rely on generalized ecological representations to simulate plant and soil processes". This means that the concepts currently expressed from line 75 onwards should appear earlier to ensure a smooth transition between topics like fine root traits, fungal collaboration gradients and DVMs.

*Thank you, but we believe the definition of the fungal collaboration gradient should in fact come after the general root trait spectrum description, and the introduction on DVMs should come close to the end.*

- Lines 78-82 could also be smoothed, e.g. "Aboveground plant traits are often analyzed using the 'leaf economics spectrum,' a framework that classifies leaves based on a trade-off between rapid growth and resource conservation (Wright et al., 2004). Including such frameworks in DVMs has provided insights into nutrient dynamics and community resilience (Sakschewski et al., 2015, 2016; Dantas de Paula et al., 2021). By contrast, belowground processes, despite their critical ecological roles, remain underrepresented in these models (Langan et al., 2017; Sakschewski et al., 2021)".

*Thanks for the suggestion, this part has been amended as follows: "Aboveground plant trait variation has been implemented using the 'leaf economics spectrum,' a framework that classifies leaves based on a trade-off between rapid growth and resource conservation (Wright et al., 2004). Including such frameworks in DVMs has provided insights into nutrient dynamics and community resilience (i.e., Sakschewski et al., 2015, 2016; Dantas de Paula et al., 2021). By contrast, belowground processes, despite their critical ecological roles, remain underrepresented in these models (Langan et al., 2017; Sakschewski et al., 2021)"*

- Similarly, to ensure a smooth transition from DVM limitations to study objectives, the authors might consider something like "Existing models simplify or omit critical variations in root traits and mycorrhizal dynamics, limiting their ability to capture site-specific belowground processes. To address these gaps, this study develops a dynamic approach that integrates detailed root trait and fungal interaction data into DVMs for broader ecological applications".

*Thank you, the last line was added to the last paragraph of the introduction, which is where our specific approach is described.*

Explaining concepts:

- The authors could further describe the "fungal collaboration gradient" (line 60) as a spectrum of strategies that plants use to interact with mycorrhizal fungi: at one end, plants invest heavily in root traits that enhance independent nutrient uptake, while at the other end they rely more on fungal partners to exchange nutrients for carbon.

*Thank you, a line related to this description was added at the end of paragraph of line 88. "A spectrum of strategies thus emerges: at one end, plants invest heavily in root traits that enhance independent nutrient uptake, while at the other end they rely more on fungal partners to exchange nutrients for carbon."*

- Similarly, the authors can briefly introduce and support DVMs as simulation tools that predict how plant communities respond to environmental changes by incorporating simplified representations of ecological processes, such as growth, nutrient cycling and competition.

***Thanks for the suggestion, but we believe the DVM concept is already explained enough here at line 91, and more details in the methods.***

Introducing hypotheses:

The current introduction does include hypotheses (from line 98 onwards), but they can be made more explicit, with a clear framing sentence following the rationale for focusing on root traits and fungal interactions. For example: "In this study, we hypothesize that incorporating the FCG into a trait-based DVM will improve predictions of root trait distributions, biomass, and productivity across nutrient gradients. Specifically, we predict that …". This approach would make the goals and hypotheses prominent and easier to follow.

***Thank you for the comment. In the last few lines of the last paragraph, the hypothesis as suggested were made more clear:***

***"We aim here to test with our model implementation the general hypothesis that the FCG is an important factor behind the observed root trait distribution, forest biomass and productivity. , More specifically, we hypothesize that (1) in line with the mycorrhizal colonization gradient and field measurements of root traits, as available nutrients decrease with elevation, simulated community average values of SRL decrease, root diameters increase and colonization rates by AMF increase when the FCG is active. Next, we removed the mycorrhizal fungi in a simulated exclusion experiment, and (2)expect that in the absence of AMF, plant biomass and productivity would be affected, and SRL between the different sites of the elevation gradient would not differ. In other words, this latter result would imply that AMF drives morphological root diversity."***

Synthesizing related studies:

While the introduction includes citations from relevant and recent studies on several topics, i.e. fine root traits (e.g. Nie et al., 2013; Bardgett et al., 2014), mycorrhizal interactions (e.g. Van Der Heijden et al., 2008; Hawkins et al., 2023), modeling approaches and limitations (e.g., Langan et al., 2017; Dantas de Paula et al., 2021), a broader review of (a) root-mycorrhizal interactions in different ecosystems (e.g., temperate or boreal forests) for comparative insights, and (b) previous attempts to incorporate

belowground traits into models (even outside of DVMs), would contextualize the novel contributions of this study.

*Thanks. More detail on different plant-mycorrhizal interactions and modelling approaches are very interesting prospects to include, and this has been done to a certain extent in the discussion – however we feel that these points should be included in further publications, where our model is extended to other environments.*

Material and methods (Section 3):

The Materials and Methods section is generally well-written, providing a solid outline of the model's structure and integration of the FCG. However, certain aspects require clarification or expansion to ensure transparency and reproducibility. Specifically, a dedicated sub-section (e.g. "3.1. Land use and vegetation cover") should be created to include detailed information about historical and current land use, vegetation types, conservation status, and anthropogenic impacts. The current description does not specify the vegetation type at each elevation site (e.g., primary forest, secondary forest, or disturbed areas), which is crucial because vegetation types significantly influence nutrient cycling, organic matter decomposition, and fungal associations. It remains unclear whether the sites are pristine or subject to human activity. For instance: Are these sites located within protected areas? Do they feature continuous forest cover, fragmented landscapes, or mixed land use? The authors should describe the dominant vegetation types or species at each elevation, such as tropical montane cloud forests, mixed forests, or other specific plant communities. A brief mention of historical land use (e.g., deforestation or reforestation efforts) would provide context for the current ecosystem conditions.

*Thank you for pointing this out. The study sites are located within a national park, and are all considered to be primary forest. This has been added to the text: "These three sites are considered to be primary forest remnants and are located within the Podocarpus National Park, where In this location, a wealth of biotic and abiotic measurements have been carried out since the early 2000's (Beck et al., 2008; Bendix et al., 2013, 2021)." Further description of vegetation types, plant community composition, land use in the surroundings, etc. are all thoroughly described in these two references and may lead to an excessive word count in the manuscript.*

This considered, the following changes are recommended to improve organization and clarity:

Section 3.1 (to be renumbered 3.2 under a new title, e.g. "Abiotic characteristics and nutrient gradients") could benefit from including quantitative data on environmental factors, such as soil

nutrient levels and moisture content, and elaborate on how these variables were incorporated into the model.

*Thank you for the suggestion, but we have already included in this section measurements of mineral nutrient content, and moisture is not a limiting factor for these three sites.*

Section 3.2 (to be renumbered 3.3) could anticipate a discussion on specific algorithms and assumptions used to simulate root growth dynamics and mycorrhizal interactions, which would add transparency to the mechanics of the model.

*Thank you for the comment, the following text has been added: "This previous implementation therefore did not provide a realistic relationship between C invested into nutrient acquisition (as roots or mycorrhizae) and effective acquisition capacity. Although root biomass in general terms translates into higher nutrient or water uptake capacity, there is high variation for same biomass values, and this is thought to result from root morphological trait variation (Kokko et al., 1993). Even though root architecture (i.e. root distribution along different soil layers) and rooting depth are considered in our and other models (Langan et al., 2017; Sakschewski et al., 2021), to our knowledge morphological traits such as root diameter and specific length have not been implemented."*

Section 3.3 (to be renumbered 3.4) could expand on the interactions between the FCG and external drivers (e.g., climate, competition, disturbances) to provide a more comprehensive view of their role in ecosystem dynamics.

*Thanks for this, the following lines were added to this section: "We expect then that in more nutrient limited environments, where C is not limiting and light is abundant, such as in higher elevation areas, mycorrhiza colonization will be high. On the other hand, in more C limited environments such as lower elevation sites (i.e. where light competition plays a large role) investment in mycorrhiza will not lead to higher fitness, leading to lower colonization rates."*

Section 3.4 (to be renumbered 3.5) could include a more detailed explanation of how the datasets used were used. This could include descriptions of the variables measured and how they were incorporated into the model to improve transparency and reproducibility.

*For this we described better the N and P deposition data, and added references for more details on how the driver data influences the results. The added text is: Soil data and parameters were taken from the World Soil database (FAO/IIASA/ISRIC/ISS-CAS/JRC, 2012); current N and P deposition rates used were from Dantas de Paula et al., (2021); which were measured weekly during the same 1999 –*

*2018 period. Details on how driver data influences ecosystem processes can be found at (Smith et al., (2014). and (Dantas de Paula et al., (2021).*

Section 3.5 (to be renumbered 3.6) could enhance the ecological relevance of the AMF-on and AMF-off scenarios by linking them to practical applications (e.g., conservation strategies and ecosystem management).

*Thank you for the comment, we believe practical applications are better suited for the discussion.*

Section 3.6 (to be renumbered 3.7) could provide a more detailed statistical analysis, including performance indicators like root mean square error (RMSE) or Nash-Sutcliffe Efficiency (EF), to assess the model accuracy.

*Thank you for this. Since both the field and simulation data have a lot of outliers, the median absolute error was used to estimate the model performance against field data. The values were added to a new table in the appendix, Table A2. In addition, a new figure was added (Figure A5) in which SRL and SLA are added as boxplots also considering the field data, in order to improve interpretation of the results. The following lines were added to section 4.2: "Model performance thus improved in the AMF-on scenario, reducing median absolute errors for SRL in comparison with the AMF-off. SLA on the other hand was slightly worse with the activation of AMF uptake (Table A2, Figure A5)."*

Section 3.7 (to be renumbered 3.8) could discuss the ecological implications of the sensitivity analysis findings, particularly highlighting their relevance to management practices and identifying areas for future research.

*Thank you for the suggestion. Please check the new additions in the discussion section for more details on this.*

Results (Section 4):

*In order to improve the transfer of this study to practical applications, a new table was added (Table 1) which includes simulated data on soil C stocks, N mineralization rates and litter stoichiometry. These new data show how the presence of AMF significantly influence soil structure and dynamics. Discussion points on this were also included.*

Section 4.1 could elaborate on the ecological significance and practical applications of the optimal rmax value (0.5). For example, how does varying rmax inform decisions on ecological restoration or nutrient cycling management?

***Thank you, this has been added in the discussion (L 518)***

Section 4.2 could provide details on the statistical significance of observed patterns (e.g., p-values) and clarify whether the AMF-off scenario exhibited an exact uniform distribution or simply a lack of observed patterns. In addition, the paragraph on SLA traits under the AMF-off scenario requires further explanation of the ecological implications. What does a "conservative" (lines 308) trait like lower SLA mean for plant strategy or ecosystem processes? Further discussion of these traits in terms of resource allocation or plant adaptation would strengthen the argument.

***Thank you. We also added that the AMF-off scenario has a lack of trends regarding SRL, and ran a t-test for the AMF scenarios along all variables, see Table 1 legend. For more details regarding this comment, please check the discussion.***

Section 4.3 requires further interpretation of the reduction in biomass at the 3,000 m site (80.6%) under the AMF-on scenarios. A brief discussion of the ecological mechanisms behind this pattern would enrich the results and help to explain why such a large reduction in biomass occurs at higher altitudes (possibly in terms of nutrient availability, plant strategy or other environmental factors).

***Please check the discussion for more details on this.***

Discussion (Section 5):

The discussion briefly touches on potential ecological mechanisms, in particular nutrient limitation and mycorrhizal requirements at higher elevations. However, there is limited discussion of the biological mechanisms driving the large biomass differences observed in the AMF-on scenario, particularly at higher elevations. As noted above, the section should expand on the biological mechanisms underlying biomass differences, particularly at higher elevations, and explore interactions between FCG and other factors (e.g., climate, competition, disturbance).

Conclusion (Section 6):

The conclusion could be strengthened by addressing interactions between the FCG and other factors, such as climate or soil type. While the authors acknowledge the need for future research on belowground traits and their interactions with AMF and aboveground traits, the discussion could be expanded to include how this understanding can be applied to ecosystem management and conservation practices.

*Thank you for suggesting this. As noted in several previous comments, links to conservation practices, management and restoration would greatly enrich the discussion. Here we have added a whole paragraph relating to this in section 5.4, which reads:*

*Finally, our study can provide important insights into the recent discussion on the importance of AMF for ecosystem management, in spite of its limitations. First, the relevance of AMF abundance and diversity for agriculture has been argued for (Rillig et al., 2019) and against (Ryan & Graham, 2002), particularly when criteria for AMF benefits are defined either as yield or sustainability. In this regard, our modelling study confirms the strong and long term influence of AMF to plant biomass and productivity for environments where nutrients are the most limiting factor, but weaker effects where they are not. Second, the shift in both SRL and SLA distributions when AMF is deactivated suggest that plant communities may differ significantly when mycorrhiza is absent, particularly towards conservative, ruderal assemblages as can be found in many degraded environments. Our model is thus in line with suggestions from ecosystem restoration practices: the introduction of AMF inoculation can both impact plant growth and community composition (Lin et al., 2015). In order to better explore from a model perspective AMF influence to the whole ecosystem, the implementation and evaluation through simulated experiments of several key processes such as soil aggregation, seedling survival, resistance to pathogens and resistance to invasive species as well as a thorough analysis of AMF effects to soil stocks and fluxes, would be invaluable (Rillig et al., 2019).*

The implications of rising atmospheric CO2 on belowground processes and mycorrhizal interactions (lines 478-479) also warrant further development.

*Thank you, a lines and reference on this were added at L 518, where AMF turnover is crucial for estimation of CO2 effects on mycorrhiza, as well as in L 538.*

In addition, exploring the potential for extending this approach to other ecosystems, such as temperate forests, grasslands or drylands, would increase the impact of the study.

*Thank you, we have added extra lines on including Ectomycorrhiza fungi and exploring this approach for estimating global patterns of mycorrhiza types.*

Finally, a more explicit link between the results of the study and broader ecological theory would strengthen the conclusions. This could include discussion of how the integration of root traits and mycorrhizal cooperation into DVMs advances our understanding of plant-soil feedback mechanisms, nutrient cycling and ecosystem resilience.

***Thanks for the suggestion, this is now explored in more detail in combination with the discussion on the soil C stocks, net nutrient mineralization and litter stoichiometry results .***

Appendix A (Section 7):

The appendix significantly enhances the transparency and reproducibility of the study by providing additional information, including data tables and analyses that help interpret the model results. However, the authors may consider omitting the heading 7.1, as it pertains to a single section.

***Thank you for the suggestion, the heading has been now omitted.***

**REVIEWER 1, 2nd comment**

The discussion of ecosystem resilience should include how the FCG contributes to resilience to environmental change (e.g. climate change, disturbance).

***Thank you for the suggestion. A direct reference for the FCG as a driver to plant resilience was not found, however the important role of AMF to plant resilience was added at the end of the above commented paragraph: "Due to their increased absorption capabilities, mycorrhiza can effectively support plant survival in harsher environments, increasing resilience to disturbances such as droughts (Das & Sarkar, 2024)."***

This reviewer pointed out the need for detailed information on vegetation type and land use in the study area. While the authors acknowledge that the sites are located within a national park and are considered primary forest, this is not sufficient. As three previous studies (Beck et al., 2008; Bendix et al., 2013, 2021) are cited, the authors should briefly summarize the key vegetation characteristics from

these studies. This could include dominant tree species, understory composition and any significant disturbances observed. This information is important for understanding the ecological context of the study and for interpreting the model results. Different vegetation types have different root architectures, mycorrhizal associations and nutrient requirements, which may affect the predictions of the model.

*In this case, the authors are happy to provide more information on the study site. A new paragraph was added in section 3.1.: "Extensive floristic inventories in the area allowed the distinction of the primary forest into four main types. Forest type I comprises a tall and species-rich forest community located in valleys at < 2,200 m.a.s.l, with trees reaching up to 35 m and dominated by the species Miconia spp. (Melastomataceae), Ocotea spp., Persea spp. (Lauraceae), Ficus spp. (Moraceae) and Inga spp. (Mimosaceae). Forest type II is located within the same altitudinal range, but along ridges, which drive the vegetation to a smaller stature (generally up to 15 m height). Common species include Alzatea verticillata Ruiz & Pav. (Alzateaceae), Graffenrieda emarginata Triana (Melastomataceae), Podocarpus oleifolius D. Don (Podocarpaceae) and Lauraceae. With increasing elevation between 2,100 and 2,250 m forest type III is found, where most trees are lower than 12 m and common taxa include Clusia cf. ducuoides Engl. (Clusiaceae), Alchornea grandiflora Müll. Arq. (Euphorbiaceae), Licaria subsessilis van der Werff (Lauraceae), Eschweilera sessilis A.C. Sm. (Lecythidaceae) and G. emarginata Triana (Melastomataceae). Finally, above 2,250 m the forest type IV is found, which is dominated by the species Purdiaea nutans Planch. (Cyrillaceae), herbs and terrestrial bromeliads (Homeier et al., 2010). Our study sites thus comprise forest types I and II (1,000 m), forest type III (2,000 m) and forest type IV (3,000 m)."*

*We have also added earlier in the section some additional information about the topography and soils, as well as main disturbances: "The terrain is predominantly steep and rugged, characterized by ridges and valleys. The primary parent materials for soil formation include Paleozoic metamorphosed schists and sandstones, interspersed with quartz veins. Cambisols are the most prevalent soil type at lower elevations, while Planosols and Histosols dominate at higher elevations around 4,230 meters (Wilcke et al., 2002). The main recurring disturbance in the area are landslides (Wilcke et al., 2003)."*

The authors acknowledge the need for further discussion of the mechanisms driving the large biomass reductions at higher altitudes in the AMF-on scenario. This reviewer encourages the authors to explore possible explanations. For example, they could clarify whether higher elevation sites are more nutrient limited. If so, they should investigate the extent to which AMF activity exacerbates this limitation, possibly linking it to changes in nutrient cycling or plant-soil feedbacks. Similarly, the authors could investigate whether increased competition for resources (light, water) at higher altitudes may contribute to the observed biomass reduction in the presence of AMF. In addition, it should be discussed whether plants at higher altitudes have different resource allocation strategies (e.g. greater investment in defence mechanisms) that may be negatively affected by AMF.

*Thank you for noting this, however we kindly ask for additional clarification, as the authors understand the presence of AMF as an alleviation (not exacerbation) factor for plant nutrient limitation – therefore the presence of AMF would contribute to the increase of biomass, as can be seen in Table 1 and Figure 4.*

The potential for extending this approach to other ecosystems may need to be explored. Although the authors mention ectomycorrhizal fungi, they could expand on the potential for applying this framework to different biomes (e.g. temperate forests, grasslands) and how the model parameters might need to be adapted for each ecosystem.

*Thank you for the suggestion – for this some sentences were added in the discussion, section 5.4: "The inclusion of EMF into DVMs should consider important differences to a AMF implementation. First, the access EMF has to organic sources requires the production of specialized enzymes, which are not present in AMF. This means that total cost borne by plants for nutrient acquisition may be higher in EMF symbiosis, as confirmed empirically (Hawkins et al., 2023). Second, kinetic uptake parameters, fungal tissue C:N ratios (which in our model impacts mycorrhiza respiration), turnover rates as well as SHL traits should differ due to distinct EMF physiology and morphology. Within the trait-varying approach, one possible implementation could be randomizing the individual's preference for AMF or EMF (or no mycorrhiza) during establishment (but not for grasslands, which are AMF exclusive)."*

**REVIEWER 1, 3rd comment**

This reviewer appreciates the authors' insights into AMF. As the previous comment regarding biomass reductions at high altitudes in the AMF-on scenario appears to have been misinterpreted, this reviewer does not suggest that AMF exacerbates nutrient limitation, but rather encourages the authors to explore how the combined effects of nutrient availability, resource competition (light, water) and resource allocation (e.g. defence) at these altitudes may influence biomass, even in the presence of AMF. Addressing these interactions in the discussion would help to strengthen the interpretation.

*Thank you for the reply, and apologies for the misinterpretation. The suggestion from the reviewer is very interesting, and indeed a point where ecosystem models can provide valuable insight. As already added by the reviewer's suggestion in section 3.3 ("We expect then that in more nutrient limited environments, such as in higher elevation areas, mycorrhiza colonization will be high. On the other hand, in lower elevation sites where light competition plays a large role, investment in mycorrhiza will not lead to higher fitness, leading to lower colonization rates."), we follow up on this with more detail further in the manuscript (section 5.1) now including a paragraph on the influence of this dynamic on biomass: "The presence of AMF in the higher elevation sites was not enough to drive the biomass in the 3,000 m sites to the same values found at 1,000 m, as observed. As evaluated in a climatic*

*sensitivity analysis in a previous LPJ-GUESS-NTD publication, the gradient of biomass and productivity in our study sites is driven ultimately by temperature impacts on nutrient cycling (Dantas de Paula et al., 2021). In the model, lower decomposition rates result is less nutrient mineralization and availability, which is exacerbated by changes in leaf and litter traits (more C in relation to N and P, tougher leaves which decompose poorly), and limit productivity and biomass. Field studies complement this, indicating that with elevation more biomass is allocated belowground suggesting higher competition for nutrients (Leuschner et al., 2013)."*

**REVIEWER 2**

I read the manuscript „The fungal collaboration gradient drives root trait distribution and ecosystem processes in a tropical montane forest" with great interest. I am not a modeler but a functional-trait ecologist, so understanding the details of the model was a bit of a challenge to me. But Figure one and specifically Figure 2 helped a lot to understand the approach. Due to my background, I will mainly comment on the traits and the ecology rather than the model itself. The integration of the collaboration gradient in an existing model is the main novelty of the paper, which is also clearly stated in the title and abstract. This aspect itself is both novel and timely and the results show very clearly that the approach was successful. I consider this an important step in model development which makes the paper a great contribution.

The results are clearly aligned and the discussion is detailed and critical. Model approaches are based on simplifying concepts and of course one could always add more nuances (like surface maximization by root hairs or the fact that the relationship between mycorrhizal colonization and extra-radical hyphal surface varies tremendously between fungal taxa). But I am satisfied with the limitations discussed here.

I can't see major flaws in the approach so all my comments are more of a minor nature:

L17: I would write fine-root rather than root only here and make sure to also clarify this at several points in the text. We know that fine-root and coarse-root traits and functions vary considerably, so being precise here is important.

*Many thanks for pointing this out. In addition to correcting broad references to roots as fine roots along the manuscript, in the first paragraph of the introduction the distinction between fine and coarse roots was briefly mentioned. Also, in the first paragraph of section 3.3, the definition of fine roots for this study (roots with less than 2 mm diameter) was added.*

L38: Rooting depth is not a fine-root trait. I see the point, that rooting depth determines where fine roots are located but it is a root-system trait.

*That is correct, thank you. The reference to rooting depth was removed.*

L48: WI guess when using the term 'global spectrum of plant form and function' one should cite the paper that coined this term: Díaz et al. 2015.

*Thank you for noting – the paper was in fact cited, but had the wrong publication year. This has been corrected.*

L63: Here it would be important to speak of fine roots as well. Specifically when citing McCormack 2015.

*Fine roots were added now here, thanks.*

L74: 'spectrum of form and function' – here again it is important to be specific. Is this a general term about plant functioning or a specific reference to the spectrum sensu Díaz?

*This is a reference to Diaz's spectrum, her reference was added, many thanks.*

L87: very transparent to start with the limitations but I would emphasize this sentence a bit more here, at least with: "Nevertheless, including the variation in root traits.... holds the potential to...". I think this statement is more than fair.

*Thank you for the suggestion. The text was now changed and reads: "Nevertheless, including variation in fine root traits and mycorrhizal colonization in DVMs holds the potential to improve predictions of vegetation response to climate change, as was the case with leaf and wood traits (Sakschewski et al., 2016)"*

L133: phrasing: The percentage of root length colonized by arbuscular mycorrhizal fungi, which interact with

*Well pointed out. The phrase now reads: "The percentage of root length colonized by arbuscular mycorrhizal fungi (AMF), which interacts with almost all species in the study area (Kottke & Haug, 2004), increases from an average of 25% at the lowest elevation to 61% at the highest (Camenzind et al., 2016)."*

L179: I don't fully get why you had to calculate diameter. GRooT should have diameter values for almost all species with SRL values, right?

***It is true that GRooT contains diameter values for species as well, however, both SRL and fine root diameter data was available at site level, therefore it was decided to use this instead.***

Figure2: Very good representation of the main story! Easy to understand for a non-modeler – thanks J

Figure5: Best figure and main result from my point of view.

---

## Author Response (AR2)

Dear Editor,

Thank you for these suggestions. I believe they are very interesting points and led to new insights on the modelling results´ implications to soil ecology.

*- An important hypothesis of your modelling work is that the plant-microbial competition for N acquisition increases and nutrient availability for plants decreases with altitude (and decreasing temperature), explaining the higher plant investment in mycorrhizal fungi. Consistently, you could dedicate some lines to explain we need more experimental works to test/support this hypothesis, including "new" methods that better qualifies nutrition status of wild plants and plant-soil competition for nutrients.*

To address this, the following text was added to section 5.1

"Field studies in our elevation gradient sites agree with our modelling results and hypothesis that plant-microbial competition for N acquisition intensifies with altitude. For instance, experimental additions of N (50 kg N ha$^{-1}$ y$^{-1}$) and P (10 kg P ha$^{-1}$ y$^{-1}$) during 5 years resulted in increased CH$_4$ uptake for the 2,000 and 3,000 m.a.s.l. sites, indicating strong nutrient competition (Martinson et al., 2021). Also, To expand on this, future studies should integrate field and laboratory experiments across different ecosystems and altitudinal gradients. Some methods offer promising avenues to improve our understanding of plant nutritional status and plant-soil competition for nutrients. For instance, stable isotope tracing (e.g., $^{15}$N labelling) can help track N uptake by plants and microbes, while enzyme activity assays can reveal microbial nutrient acquisition strategies under varying temperature conditions (Dunn et al., 2006). Additionally, soil sterilization bioassays can offer insights into the role of soil microbes affecting plant performance (Waring et al., 2016), whilst metagenomics and metatranscriptomics can provide insights into shifts in microbial communities and their functional roles in nutrient cycling (Mendes et al., 2017). Expanding empirical research in these areas will be essential to refine our understanding of plant-microbial interactions in response to changing environmental conditions."

*- Your discussions omit the key role of free microbial decomposers in the supply of soluble nutrients for plants, including those suppling nutrients from evolved organic matter and mineral associated organic matter (e.g. https://doi.org/10.1111/gcb.17034; https://doi.org/10.1111/1365-2435.14038). Try to better balance your discussions by adding some lines on the role of these microbes.*

Thank you, this paragraph was added to the section 5.1

"While microbial competition for nutrients can limit plant nutrient acquisition, free-living microbial decomposers play a crucial role in breaking down evolved organic matter and mineral-associated organic matter, thereby releasing soluble nutrients to plants synchronizing supply and demand (Fontaine et al., 2024). These microbes, including saprotrophic fungi and bacteria, drive nutrient mineralization processes that can enhance plant nutrient uptake, particularly in nutrient-limited

environments such as our studied elevation gradient. In our sites, the N cycle is closely coupled (i.e., gross N mineralization is equal to $NH_4$ immobilization, and gross nitrification is equal to $NO_3$ immobilization), and experimental nutrient additions can alter this equilibrium (Baldos et al., 2015). This is particularly alarming due to the observed increase of anthropogenic N deposition in these areas (Wilcke et al., 2013). Other global environmental changes such as $CO_2$ and temperature increase may impact plant controls on soil organic matter dynamics which are mediated by microbes through priming effects, leading to the loss of nutrients and ecosystem degradation (Bernard et al., 2022). Balancing the interactions between decomposer activity and plant-microbial competition is, therefore, essential for understanding nutrient dynamics along environmental gradients."

I hope the new additions were satisfactory and provide new clarity into the implication of the results.

Best regards,

Mateus Dantas de Paula, in behalf of all authors

References

Baldos, A. P., Corre, M. D., and Veldkamp, E.: Response of N cycling to nutrient inputs in forest soils across a 1000-3000 m elevation gradient in the Ecuadorian Andes, Ecology, 96, 749–761, https://doi.org/10.1890/14-0295.1, 2015.

Bernard, L., Basile-Doelsch, I., Derrien, D., Fanin, N., Fontaine, S., Guenet, B., Karimi, B., Marsden, C., and Maron, P. A.: Advancing the mechanistic understanding of the priming effect on soil organic matter mineralisation, Funct. Ecol., 36, 1355–1377, https://doi.org/10.1111/1365-2435.14038, 2022.

Dunn, R. M., Mikola, J., Bol, R., and Bardgett, R. D.: Influence of microbial activity on plant-microbial competition for organic and inorganic nitrogen, Plant Soil, 289, 321–334, https://doi.org/10.1007/s11104-006-9142-z, 2006.

Fontaine, S., Abbadie, L., Aubert, M., Barot, S., Bloor, J. M. G., Derrien, D., Duchene, O., Gross, N., Henneron, L., Le Roux, X., Loeuille, N., Michel, J., Recous, S., Wipf, D., and Alvarez, G.: Plant–soil synchrony in nutrient cycles: Learning from ecosystems to design sustainable agrosystems, Glob. Chang. Biol., 30, 1–24, https://doi.org/10.1111/gcb.17034, 2024.

Martinson, G. O., Müller, A. K., Matson, A. L., Corre, M. D., and Veldkamp, E.: Nitrogen and Phosphorus Control Soil Methane Uptake in Tropical Montane Forests, J. Geophys. Res. Biogeosciences, 126, 1–14, https://doi.org/10.1029/2020JG005970, 2021.

Mendes, L. W., Braga, L. P. P., Navarrete, A. A., Souza, D. G. de, Silva, G. G. Z., and Tsai, S. M.: Using Metagenomics to Connect Microbial Community Biodiversity and Functions, Curr. Issues Mol. Biol., 24, 103–118, https://doi.org/10.21775/CIMB.024.103, 2017.

Waring, B. G., Gei, M. G., Rosenthal, L., and Powers, J. S.: Plant-microbe interactions along a gradient of soil fertility in tropical dry forest, J. Trop. Ecol., 32, 314–323, https://doi.org/10.1017/S0266467416000286, 2016.

Wilcke, W., Leimer, S., Peters, T., Emck, P., Rollenbeck, R., Trachte, K., Valarezo, C., and Bendix, J.: The nitrogen cycle of tropical montane forest in Ecuador turns inorganic under environmental change, Global Biogeochem. Cycles, 27, 1194–1204, https://doi.org/10.1002/2012GB004471, 2013.